# Modular Properties of Generalised Gibbs Ensembles

**Max Downing[1]⋆ and Faisal Karimi[2]†**

**1** Laboratoire de Physique de l'École Normale Supérieure,
ENS, Université PSL, CNRS, Sorbonne Université,
Université Paris Cité, F-75005 Paris, France
**2** Department of Mathematics, King's College London,
Strand, London, WC2R 2LS, United Kingdom

⋆ max.downing@phys.ens.fr , † faisal.karimi@kcl.ac.uk

## Abstract

We investigate the modular properties of Generalised Gibbs Ensembles (GGEs) in two dimensional conformal field theories. These are obtained by inserting higher spin charges in the expressions for the partition function of the theory. We investigate the particular case where KdV charges are inserted in the GGE. We first determine an asymptotic expression for the transformed GGE. This expression is an expansion in terms of the zero modes of all the quasi-primary fields in the theory, not just the KdV charges. While these charges are non-commuting they can be re-exponentiated to give an asymptotic expression for the transformed GGE in terms of another GGE. As an explicit example we focus on the Lee-Yang model. We use the Thermodynamic Bethe Ansatz in the Lee-Yang model to first replicate the asymptotic results, and then find additional energies that need to be included in the transformed GGE in order to find the exact modular transformation.

## Contents

---

# 1   Introduction

The study of generalised Gibbs ensembles plays an important role in understanding the thermalisation properties of many body systems with additional conserved quantities. Usually when we study a system where the only conserved quantity is the energy we use the Gibbs distribution

$$p_n = \frac{1}{Z} e^{-\beta E_n} , \quad Z = \sum_n e^{-\beta E_n} , \tag{1}$$

which gives the probability of the system being in state $n$ which has energy $E_n$. However if we are interested in a system which contains additional conserved charges $Q_i$, not just the energy, we instead use the generalised Gibbs distribution

$$p_n = \frac{1}{Z} e^{-\beta E_n - \sum_i \alpha_i Q_{i,n}} , \quad Z = \sum_n e^{-\beta E_n - \sum_i \alpha_i Q_{i,n}} , \tag{2}$$

where $Q_{i,n}$ is the value of the charge $Q_i$ in state $n$. For a review of the role of GGEs in the contexts of statistical mechanics and thermalisation see [1].

In this paper we will be interested in GGEs in two dimensional conformal field theories (2d CFTs). In order to construct a GGE we need to have additional conserved charges. To construct these charges in a 2d CFT we start with a quasi-primary field. These fields give rise to all the conserved charges in the theory. We will be interested in the modular properties of the GGE and hence we want to study theories on a torus. For us the torus will be a cylinder with the ends identified and therefore our charges will be the zero modes of the quasi-primary fields on a cylinder.

Often it is not enough for our theory to have an infinite set of conserved charges, we also want the charges to commute. It has been known for some time that 2d CFTs contain infinite sets of mutually commuting conserved charges [2]. The most well known set of charges are related to the classical KdV hierarchy, as detailed in [3], and hence we will refer to them as the KdV charges. They are constructed from the Virasoro modes and we list the first three here

$$I_1(R) = \frac{2\pi}{R}\left(L_0 - \frac{c}{24}\right) , \tag{3}$$

$$I_3(R) = \left(\frac{2\pi}{R}\right)^3 \left(2\sum_{k=1}^{\infty} L_{-k}L_k + L_0^2 - \frac{c+2}{12}L_0 + \frac{c(5c+22)}{2880}\right) , \tag{4}$$

$$I_5(R) = \left(\frac{2\pi}{R}\right)^5 \left(\sum_{k_1+k_2+k_3=0} :L_{k_1}L_{k_2}L_{k_3}: + \sum_{k=1}^{\infty}\left(\frac{c+11}{6}k^2 - 1 - \frac{c}{24}\right)L_{-k}L_k \right. \tag{5}$$

$$\left. +\frac{3}{2}\sum_{k=1}^{\infty} L_{1-2k}L_{2k-1} - \frac{c+4}{8}L_0^2 + \frac{(c+2)(3c+20)}{576}L_0 - \frac{c(3c+14)(7c+68)}{290304}\right) .$$

The normal ordering $:L_{k_1}L_{k_2}L_{k_3}:$ means we order the modes such that $k_1 \leq k_2 \leq k_3$. These charges are the zero modes of quasi-primary fields on a cylinder of circumference $R$. Often in the literature the dimensionless charges $I_{2n-1} = \left(\frac{R}{2\pi}\right)^{2n-1} I_{2n-1}(R)$ are studied. Informally the charges $I_{2n-1}$ are given by $I_{2n-1}(2\pi)$, i.e. the charges defined on a cylinder with $R = 2\pi$, and hence the prefactor is absent. However we note that $R$ is dimensionful and therefore cannot actually be set to the dimensionless quantity $2\pi$. For our purposes this prefactor will play an important role and hence we will keep it explicit.

These are the additional conserved charges that we will insert into our partition functions to obtain a GGE

$$Z = \text{Tr}\left(e^{-L(I_1(R) + \sum_{n=2}^{\infty}\alpha_{2n-1}I_{2n-1}(R))}\right) , \tag{6}$$

where $L$ is the length of the cylinder. At this stage we have not been explicit about what space we are tracing over, it could be individual highest weight representations of the Virasoro algebra or the whole space of states. Later we will explicitly be tracing over individual highest weight representations.

These GGEs have been studied extensively in the literature. Their large central charge limit ($c \to \infty$) was studied in a series of papers by Dymarsky $et$ $al$ [4–6] and also by Maloney $et$ $al$ in [7] and Brehm and Das in [8]. There, expressions for these GGEs in the limit $c \to \infty$ and leading $1/c$ corrections were derived. These GGEs are then holographically dual to a class of black holes in AdS$_3$ referred to in [9] as KdV charged black holes and their connection to the eigenstate thermalisation hypothesis was also explored in [10].

In this paper we will be investigating the modular properties of these GGEs. The deep relationship between 2d CFTs and modular forms has been known for a long time and has been used extensively to study 2d CFTs. It is know that the characters of a rational 2d CFT form a vector valued modular form. This was first suggested by Cardy in [11] and rigorously proven by Zhu in [12]. It is also known that if we expand the GGE (6) as a power series in chemical potentials $\alpha_{2n-1}$, then each term, which is a correlation functions of the charges, is a modular form or quasi-modular form. This was first argued by Dijkgraaf in [13] and then in [14] Maloney $et$ $al$ found expressions for the correlators in terms of modular differential operators acting on the characters which makes the modular properties manifest.

A natural question to ask is whether the full GGE has any interesting modular properties. This has been studied in detail for the GGEs in the free fermion model ($c = \frac{1}{2}$ Ising minimal model) in the series of papers [15–17]. In general, closed form expressions for the GGEs are not known. However for the free fermion model the simplicity of the theory means that exact

expressions can be found and used as a starting point to study the modular properties. This meant an explicit expression for the modular transformation could be found.

In order to find this modular transformation formula, first the GGEs were expanded as an asymptotic power series in the chemical potentials. Each term in the series could be modular transformed and then the result was resummed into an exponential. This gave another GGE that contained an infinite set of charges, however this expression diverged and had to be regularised. Even after regularising the result, the expressions only matched asymptotically which is not surprising since in the first step of the derivation we take an asymptotic expansion.

However an exact modular transform can be found. This was done by using the thermodynamic Bethe ansatz (TBA). The TBA for the original GGE is known from [18]. We then take a mirror transform of the TBA in order to find the spectrum of the GGE in the new channel. When this is done the spectrum from the asymptotic results can be reproduced, but we also find additional energies. These energies behave as $C\alpha^{-\nu}$, where $\alpha < 0$ is the chemical potential, $\mathrm{Re}(C) > 0$ and $\nu > 0$, and hence when they are exponentiated they give rise to terms that have a vanishing asymptotic expansion. This is why they were missed in the original asymptotic analysis but including them in the transformation gives an exact expression for the modular transformed GGE.

In this paper we want to find the modular transformation of other minimal models. We will start by briefly discussing generic minimal models and then move to focusing on the Lee-Yang minimal model. We have chosen the Lee-Yang model as our main example since the two characters satisfy a second order modular differential equation which simplifies the correlators in the asymptotic expansion and later when we solve the TBA equations there is only one integral equation to solve.

The layout of the paper is as follows. In section 3 we consider a generic rational 2d CFT and start by asymptotically expanding the GGE as a power series in the chemical potential. Each term in the series can be written as a modular differential operator acting on the characters of the theory. We can modular transform each term in the series. After taking the modular transform the resulting expressions can be written as the correlators of charges from all quasi-primary fields in the theory. We find conditions under which this restricts to just the KdV charges.

In section 4 we repeat the above asymptotic analysis for the Lee-Yang model. We again find that additional charges, not just the KdV charges, will appear in our expression. However the transformed expression can still be re-exponentiated to give an asymptotic expression for the modular transform of the GGE in terms of another GGE, this time containing all charges from quasi-primary fields, not just the KdV charges. These additional charges don't commute, and so it is not obvious that the expression will exponentiate. However we show that this does not stop us from being able to re-exponentiate the expression (up to the order, in the chemical potential $\alpha$, we are working).

We then turn our attention to the TBA in section 5. We start by using the TBA to reproduce the asymptotic results. We then show that there are other solutions to the TBA equations which when exponentiated have a vanishing asymptotic expansion, just as in the case of the free fermion model. We conjecture that including these additional energies in the transformed GGE will give the exact modular transformation for the GGE. During this process we derive new integral equations that encode the spectrum of the KdV charges as well as the charges coming from the other quasi-primary fields in the theory.

We end with a summary of the main results in section 6 and discuss some future directions. We have also included a series of appendices that contain either background material or lengthy calculations that would have cluttered the main text.

## 137 **2 Transformed GGEs and defects**

138 We start by outlining the aims of this paper. Our goal is to understand how to take a modular
139 transformation of a generalised Gibbs ensemble (GGE) in a 2d CFT. As will be explained below,
140 our CFT is living on a cylinder with the two ends identified. The GGE is given by inserting
141 a defect that wraps the compact direction of the cylinder. A modular transformation then
142 corresponds to rotating the defect so it now runs along the axis of the cylinder. The defect is
143 now intersecting the circle that the Hilbert space is defined on which leads to a new defect
144 Hilbert space and a defect Hamiltonian that acts on this space. In order to determine the
145 modular transformed GGE we need to compute this defect Hilbert space and Hamiltonian.

146 The objects we will be studying are GGEs where the additional charges inserted in the
147 characters are the KdV charges $I_{2n-1}(R)$. We will restrict ourselves to the case where we have
148 just one KdV charge inserted along with the usual 2d CFT Hamiltonian $I_1(R) = \frac{2\pi}{R}\left(L_0 - \frac{c}{24}\right)$

$$\mathrm{Tr}_{\mathcal{H}_i}\left(e^{-L(I_1(R)+\alpha I_{2n-1}(R))}\right)\,, \tag{7}$$

149 where $\mathcal{H}_i$ is a highest weight irreducible representation of the Virasoro algebra.

150 Let $\{|m\rangle\}$ be an orthonormal basis of states for the representation $\mathcal{H}_i$. By construction all
151 of the KdV charges commute, hence we can find a basis where each element is an eigenstate
152 of the charges $I_{2n-1}(R)$. The basis element $|m\rangle$ has eigenvalue $E_m^{(2n-1)}(R)$ under the charge
153 $I_{2n-1}(R)$, i.e.

$$I_{2n-1}(R)|m\rangle = E_m^{(2n-1)}(R)|m\rangle\,. \tag{8}$$

154 The GGE (7) then has the explicit form

$$\mathrm{Tr}_{\mathcal{H}_i}\left(e^{-L(I_1(R)+\alpha I_{2n-1}(R))}\right) = \sum_m e^{-L\left(E_m^{(1)}(R)+\alpha E_m^{(2n-1)}(R)\right)}\,. \tag{9}$$

155 Throughout the paper we will refer to the terms $E_m(R) = E_m^{(1)}(R) + \alpha E_m^{(2n-1)}(R)$, in the expo-
156 nential, as the spectrum of the GGE.

157 The GGE can be thought of as the insertion of a defect as was done in [16]. We consider
158 our theory to be living on a cylinder of circumference $R$ and length $L$ as shown in diagram (I)
159 of figure 1. We identify the ends of the cylinder so it becomes a torus with modular parameter
160 $\hat{\tau} = iL/R$. The insertion of the KdV charge $I_{2n-1}$ is given by a horizontal defect wrapping the
161 cylinder. The defect operator is $\hat{D} = e^{-L\alpha I_{2n-1}(R)}$ and the GGE is given by

$$\mathrm{Tr}_{\mathcal{H}_i}\left(e^{-L(I_1(R)+\alpha I_{2n-1}(R))}\right) = \mathrm{Tr}_{\mathcal{H}_i}\left(\hat{D}e^{-LI_1(R)}\right)\,. \tag{10}$$

162 The insertion of the defect doesn't change the Hilbert space we trace over but it does change
163 the spectrum of our GGE.

164 We want to take the modular transformation of the GGE (7). We will just focus on the $S$
165 transform $S: \hat{\tau} \mapsto \tau = -1/\hat{\tau}$. This is equivalent to rotating the cylinder as is shown in diagram
166 (II) of figure 1. The modular parameter becomes

$$\hat{\tau} = iL/R \mapsto \tau = -1/\hat{\tau} = iR/L\,. \tag{11}$$

167 We are now considering our theory as depicted in (II) of figure 1. The Hilbert space now lives
168 on a horizontal slice of length $L$. This horizontal slice is intersected by our defect which has
169 been rotated to be vertical. Since the defect is not topological, the resulting Hilbert space does
170 not have to carry an action of the Virasoro algebra. We denote this modified Hilbert space by
171 $\mathcal{H}_D$. The transformed GGE takes the form

$$\mathrm{Tr}_{\mathcal{H}_D}\left(e^{-RH_D(L)}\right)\,, \tag{12}$$

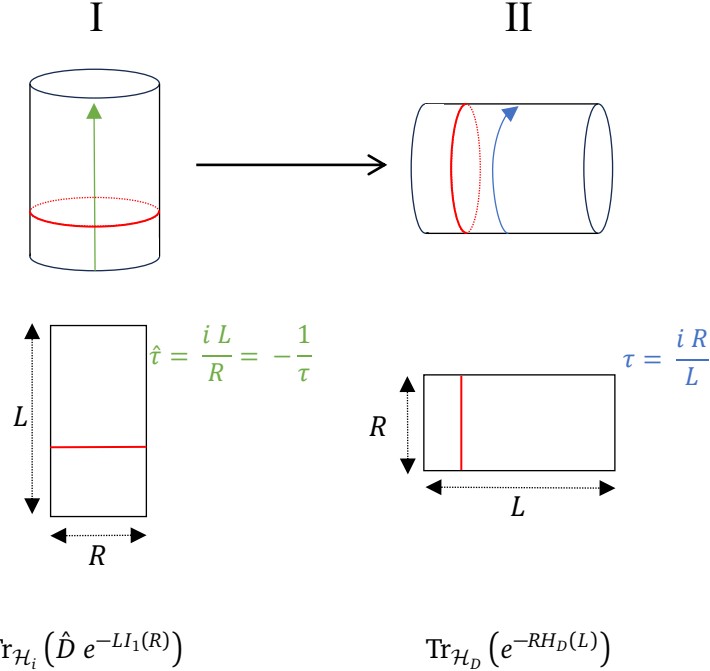

I        II

$$\hat{\tau} = \frac{i\,L}{R} = -\frac{1}{\tau}$$

$$\tau = \frac{i\,R}{L}$$

$$\mathrm{Tr}_{\mathcal{H}_i}\left(\hat{D}\,e^{-LI_1(R)}\right) \qquad\qquad \mathrm{Tr}_{\mathcal{H}_D}\left(e^{-RH_D(L)}\right)$$

Figure 1: Interpretation of the modular transformed GGE traces: on torus (I), the GGE is given by a defect inserted as an operator $\hat{D}$ in the trace; on torus (II) the defect is rotated and the transformed GGE is given by a trace over the Hilbert space $\mathcal{H}_D$ with a defect Hamiltonian $H_D(L)$ inserted in the trace.

where $H_D(L)$ is the Hamiltonian that acts on the Hilbert space $\mathcal{H}_D$.

Let $\{|\widetilde{m}\rangle\}$ be a basis for the Hilbert space $\mathcal{H}_D$ such that the element $|\widetilde{m}\rangle$ has eigenvalue $E_{\widetilde{m}}^{(D)}(L)$ under $H_D(L)$, i.e.

$$H_D(L)|\widetilde{m}\rangle = E_{\widetilde{m}}^{(D)}(L)|\widetilde{m}\rangle \,. \tag{13}$$

We can then express our transformed GGE as the sum

$$\mathrm{Tr}_{\mathcal{H}_D}\left(e^{-RH_D(L)}\right) = \sum_{\widetilde{m}} e^{-RE_{\widetilde{m}}^{(D)}(L)} \,. \tag{14}$$

We will refer to the terms $E_{\widetilde{m}}^{(D)}(L)$ as the transformed spectrum.

When $\alpha = 0$ the defect isn't present and the GGEs (7) are the characters of the 2d CFT. It is known that the characters form vector valued modular forms [12]

$$\mathrm{Tr}_{\mathcal{H}_i}\left(e^{-LI_1(R)}\right) = \sum_j S_{ij}\,\mathrm{Tr}_{\mathcal{H}_j}\left(e^{-RI_1(L)}\right) \,, \tag{15}$$

for a constant matrix $S_{ij}$. When $\alpha \neq 0$ and the defect is present we want to determine whether, under a modular transformation, the GGE (7) transforms in an analogous way to the characters in (15)

$$\mathrm{Tr}_{\mathcal{H}_i}\left(e^{-L(I_1(R)+\alpha I_{2n-1}(R))}\right) = \sum_j S_{ij}\,\mathrm{Tr}_{\mathcal{H}_{D,j}}\left(e^{-RH_D(L)}\right) \,, \tag{16}$$

where the $\mathcal{H}_{D,j}$ are a collection of defect Hilbert spaces. Or equivalently using (9) and (14)

$$\sum_m e^{-L\left(E_m^{(1)}(R)+\alpha E_m^{(2n-1)}(R)\right)} = \sum_j S_{ij}\sum_{\widetilde{m}} e^{-RE_{\widetilde{m}}^{D,j}(L)} \,, \tag{17}$$

where $E_{\tilde{m}}^{D,j}(L)$ is the spectrum of the Hamiltonian $H_D(L)$ acting on the Hilbert space $\mathcal{H}_{D,j}$.

If we take the full partition function of a 2d CFT, with both holomorphic and anti-holomorphic sectors, then physically we expect it to be modular invariant. When $\alpha = 0$ the full partition function is

$$Z(R, L) = \sum_{ij} M_{ij} \operatorname{Tr}_{\mathcal{H}_i} \left( e^{-LI_1(R)} \right) \operatorname{Tr}_{\bar{\mathcal{H}}_j} \left( e^{-L\bar{I}_1(R)} \right) , \tag{18}$$

where $\bar{\mathcal{H}}_j$ is an irreducible representation of the anti-holomorphic Virasoro algebra $\{\bar{L}_n\}$, the constants $M_{ij}$ are non-negative integers and $\bar{I}_1(R) = \frac{2\pi}{R}(\bar{L}_0 - \frac{c}{24})$. (We are assuming that the holomorphic and anti-holomorphic sectors have the same central charge.) Under the modular transformation (15), the partition function is modular invariant ($Z(R, L) = Z(L, R)$) provided the matrix $M_{ij}$ satisfies

$$M_{ij} = \sum_{kl} S_{ik} \bar{S}_{jl} M_{kl} , \tag{19}$$

where $\bar{S}_{jl}$ is the complex conjugate of $S_{jl}$. We now define the GGE of the full theory by summing over both holomorphic and anti-holomorphic sectors. We will only insert a charge in the holomorphic sector, so our GGE is

$$Z(R, L, \alpha) = \sum_{ij} M_{ij} \operatorname{Tr}_{\mathcal{H}_i} \left( e^{-L(I_1(R) + \alpha I_{2n-1}(R))} \right) \operatorname{Tr}_{\bar{\mathcal{H}}_j} \left( e^{-L\bar{I}_1(R)} \right) . \tag{20}$$

If we assume that the modular transformation (16) holds then modular invariance of the GGE (20) is given by

$$Z(R, L, \alpha) = \sum_{ij} M_{ij} \operatorname{Tr}_{\mathcal{H}_{D,i}} \left( e^{-RH_D(L)} \right) \operatorname{Tr}_{\bar{\mathcal{H}}_j} \left( e^{-R\bar{I}_1(L)} \right) , \tag{21}$$

where we used (19). Note that the $\alpha$ dependence of the transformed GGE (21) is in both the defect Hilbert spaces $\mathcal{H}_{D,i}$ and the defect Hamiltonian $H_D(L)$.

We want to determine the defect Hilbert space $\mathcal{H}_D$ of the transformed GGE and the Hamiltonian $H_D(L)$ that acts on this space. In order to try and determine the Hilbert space $\mathcal{H}_D$ and the Hamiltonian $H_D(L)$ we will make some assumptions about their form.

We will start with an asymptotic analysis, as $\alpha \to 0$, of the modular transformation of (7) in sections 3 and 4. There, as was also done in the asymptotic analysis in [15], we will assume that the defect Hilbert space is just the irreducible representations of the Virasoro algebra. In [15], where the free fermion model was studied, it was found that the defect Hamiltonian $H_D(L)$ had an asymptotic expansion as a sum over the other KdV charges

$$H_D(L) \sim \sum_{n=1}^{\infty} \alpha_{2n-1} I_{2n-1}(L) , \tag{22}$$

where $\alpha_{2n-1}$ were coefficients that depended on $\alpha$ but not $R$ and $L$.

We will see in section 3 for a generic CFT (and in section 4 for the Lee-Yang model) that this is no longer true. Instead in a generic CFT it appears that the asymptotic expansion takes the form

$$H_D(L) \sim \sum_{n=1}^{\infty} \sum_a \beta_{2n-1}^a J_{2n-1}^{(a)}(L) , \tag{23}$$

where the charges $J_{2n-1}^{(a)}$ are all the charges coming from the quasi-primary fields at level $2n$, not just the KdV charges. More details are given in section 3.

While we have an asymptotic expression for the Hamiltonian $H_D(L)$, based on the results in [15–17] we believe this is not the full picture. There, additional terms had to be added to the transformed spectrum that behaved as $\alpha^{-\nu}$ for $\nu > 0$. These terms were missed in the

asymptotic analysis since the exponential $e^{-\alpha^{-\nu}}$ has a vanishing asymptotic expansion as $\alpha \to 0$ from above. These additional terms are found in section 5.6 where the power $\nu$ is derived.

These additional terms that needed to be added to the transformed spectrum were determined in [15, 16] by using the thermodynamic Bethe ansatz (TBA). In section 5 we again use the TBA to find additional terms that we believe should be added to the transformed spectrum in order to give the full modular transformed GGE (12). These additional terms in the spectrum come from additional terms that have been added to the Hilbert space $\mathcal{H}_D$, hence this Hilbert space is no longer an irreducible representation of the Virasoro algebra.

## 3  GGEs in a Generic 2d CFT

We start by considering GGEs with a KdV charge inserted for a generic 2d CFT. For simplicity we will just consider inserting a single charge but this will already lead to interesting results. As was done in [15], we will start by expanding the GGE as an asymptotic series in the chemical potential associated to the inserted charge. We can then modular transform each term using the results from [14]. When this was done in [15] for the free fermion model ($c = \frac{1}{2}$ Ising minimal model) we found that the transformed expressions could be written as correlators of the other KdV charges. In the case of a generic CFT we will find that the transformed expressions are instead given by correlators of all the charges from quasi-primary fields, not just the KdV charges.

We will assume that we are working with a minimal model so we have a finite number of highest weight, irreducible representations of the Virasoro algebra, $\mathcal{H}_i$, whose weights are denoted by $h_i$, $i = 1, \ldots, N$. We will first consider the simplest case: a GGE with just the $I_3(R)$ charge from (4) inserted. The GGE in the $h_i$ representation $\mathcal{H}_i$ is given by

$$\mathrm{Tr}_{\mathcal{H}_i}\left(e^{-L(\alpha I_3(R) + I_1(R))}\right). \tag{24}$$

We will begin by expanding the GGE as an asymptotic series in the chemical potential $\alpha$

$$\mathrm{Tr}_{\mathcal{H}_i}\left(e^{-L(\alpha I_3(R) + I_1(R))}\right) = \sum_{n=0}^{\infty} \frac{(-\alpha L)^n}{n!} \mathrm{Tr}_{\mathcal{H}_i}\left(I_3(R)^n e^{-L I_1(R)}\right). \tag{25}$$

We can take a modular transform for each term and attempt to resum them to give us an asymptotic expression for the transformed GGE. We start by introducing the following notation: $I_{2n-1} = \left(\frac{R}{2\pi}\right)^{2n-1} I_{2n-1}(R)$, $\hat{\tau} = iL/R$ is the modular parameter of the torus and $\hat{q} = e^{2\pi i \hat{\tau}}$. We also introduce the expectation value for an operator $\mathcal{O}$

$$\langle \mathcal{O} \rangle_i(\hat{\tau}) = \mathrm{Tr}_{\mathcal{H}_i}\left(\mathcal{O} \hat{q}^{I_1}\right). \tag{26}$$

The asymptotic expansion (25) becomes

$$\mathrm{Tr}_{\mathcal{H}_i}\left(e^{-L(\alpha I_3(R) + I_1(R))}\right) = \sum_{n=0}^{\infty} \frac{1}{n!}\left(\frac{-(2\pi)^3 \alpha L}{R^3}\right)^n \langle I_3^n \rangle_i(\hat{\tau}). \tag{27}$$

The modular properties of the thermal correlators $\langle I_3^n \rangle_i$ where studied by A. Maloney *et al* in [14]. There they showed the correlators can be written as modular linear differential operators acting on the characters of the CFT. In particular up to order $\alpha^2$ we have the following

expressions for the correlators

$$\langle 1 \rangle_i = \chi_i \,, \tag{28}$$

$$\langle I_3 \rangle_i = \left( D^2 + \frac{c}{1440} E_4 \right) \chi_i \,, \tag{29}$$

$$\langle I_3^2 \rangle_i = \left( D^4 + \frac{c+40}{720} E_4 D^2 - \frac{3c+11}{1080} E_6 D + \frac{c(407c+4000)}{14515200} E_4^2 \right) \chi_i \tag{30}$$

$$+ E_2 \left( \frac{2}{3} D^3 + \frac{3c+11}{1080} E_4 D - \frac{c(c+10)}{36288} E_6 \right) \chi_i \,,$$

where $\chi_i = \chi_i(\hat{q})$ is the character of the $\mathcal{H}_i$ representation. The differential operators are given by $D^n = D_{2n-2} D_{2n-4} \dots D_0$ where $D_r$ is the Serre derivative

$$D_r = \hat{q} \frac{\partial}{\partial \hat{q}} - \frac{r}{12} E_2(\hat{\tau}) \,, \tag{31}$$

and $E_{2k}$ are the Eisenstein series defined in appendix A.

We now want to take the modular transform of each term in the asymptotic expansion of the GGE. We will just take the $S : \hat{\tau} \mapsto \tau = -1/\hat{\tau}$ transform. The characters (28) of a 2d CFT form a weight 0 vector valued modular form [12], so under the $S$ modular transform we have

$$\chi_i(\hat{\tau}) = \sum_{j=1}^{N} S_{ij} \chi_j(\tau) \,, \tag{32}$$

for a constant matrix $S_{ij}$. We can use the modular properties of Eisenstein series and Serre derivatives (given in appendix A) to compute the modular transform of the higher correlators. The one point function (29) is a weight 4 vector valued modular form

$$\langle I_3 \rangle_i(\hat{\tau}) = \tau^4 \sum_{j=1}^{N} S_{ij} \langle I_3 \rangle_j(\tau) \,. \tag{33}$$

The 2 point correlator transforms as a weight 8, depth 1 vector valued quasi-modular form

$$\langle I_3^2 \rangle_i(\hat{\tau}) = \sum_{j=1}^{N} S_{ij} \left( \tau^8 \langle I_3^2 \rangle_j(\tau) - \frac{i\tau^7}{\pi} \left( 4D^3 + \frac{3c+11}{180} E_4 D - \frac{c(c+10)}{6048} E_6 \right) \chi_j \right) \,. \tag{34}$$

The definition of quasi-modular forms is again given in appendix A.

The additional term in the transformation (34)

$$\left( 4D^3 + \frac{3c+11}{180} E_4 D - \frac{c(c+10)}{6048} E_6 \right) \chi_j \,, \tag{35}$$

can be interpreted as the thermal correlator of a linear combination of a charge $J_5$ and the KdV charge $I_5$. The charge is $J_5 = J_5(2\pi)$ where $J_5(R)$ is given by

$$J_5(R) = \left( \frac{2\pi}{R} \right)^5 \left( -\frac{18}{5} \sum_{k=1}^{\infty} k^2 L_{-k} L_k - \frac{3}{100} L_0 + \frac{31c}{16800} \right) . \tag{36}$$

This is the zero mode, on the cylinder, of a quasi-primary field at level 6 that is linearly independent to the KdV charge $I_5$. We show how to compute this charge in appendix B.2. Using the differential operator representation of the thermal correlators from appendix C.2 we find

$$4\langle I_5 \rangle_j + \frac{5}{54}(c+2)\langle J_5 \rangle_j = \left( 4D^3 + \frac{3c+11}{180} E_4 D - \frac{c(c+10)}{6048} E_6 \right) \chi_j \,. \tag{37}$$

Recalling that $\hat{\tau} = iL/R$, and hence $\tau = iR/L$, we can express the modular transformations (32–34) as

$$\langle 1 \rangle_i(\hat{\tau}) = \sum_{j=1}^{N} S_{ij} \langle 1 \rangle_j(\tau) \,, \tag{38}$$

$$\langle I_3(R) \rangle_i(\hat{\tau}) = \frac{R}{L} \sum_{j=1}^{N} S_{ij} \langle I_3(L) \rangle_j(\tau) \,, \tag{39}$$

$$\langle I_3(R)^2 \rangle_i(\hat{\tau}) = \left(\frac{R}{L}\right)^2 \sum_{j=1}^{N} S_{ij} \left( \langle I_3(L)^2 \rangle_j(\tau) - \frac{1}{R} \left( 8 \langle I_5(L) \rangle_j(\tau) + \frac{5(c+2)}{27} \langle J_5(L) \rangle_j(\tau) \right) \right). \tag{40}$$

If we assume the transformed GGE can be resummed into an exponential, we have

$$\mathrm{Tr}_{\mathcal{H}_i} \left( e^{-L(\alpha I_3(R) + I_1(R))} \right) \sim \sum_{j=1}^{N} S_{ij} \, \mathrm{Tr}_{\mathcal{H}_j} \left( e^{-R(I_1(L) + \alpha I_3(L) + \alpha^2(8 I_5(L) + \frac{5(c+2)}{27} J_5(L)) + \dots)} \right). \tag{41}$$

We have written this as a trace again to make it explicit that the right hand side can be formally interpreted as a Hamiltonian acting on a Hilbert space of states defined on a circle of circumference $L$.

Here we have assumed that after taking the modular transform of each term in (25) we can resum it into an exponential. However the charge $J_5$ doesn't commute with the KdV charges and hence we need to be careful about the order of the operators when we expand the exponential. When we study the GGE in the Lee-Yang model in the next section we will verify that the asymptotic expansion can indeed be resummed into an exponential after transforming each term.

We can see that generically when we want to take the modular transform of a GGE with a KdV charge inserted we have to include all possible charges in the transformed GGE, not just the original KdV charges.

Let us outline what will happen at higher orders in the asymptotic expansion. We will also consider the case with just one charge inserted again, but this time insert the $I_{2m-1}(R)$ charge. Hence we want to study the GGE

$$\mathrm{Tr}_{\mathcal{H}_i} \left( e^{-L(I_1(R) + \alpha I_{2m-1}(R))} \right). \tag{42}$$

If we again expand the GGE as an asymptotic series in $\alpha$ each term is of the form

$$\langle I_{2m-1}^n \rangle_i(\hat{\tau}) \,, \tag{43}$$

where we have removed the $R$ and $L$ dependence. As a function of $\hat{\tau}$, $\langle I_{2m-1}^n \rangle_i(\hat{\tau})$ is a vector valued quasi-modular form of weight $2mn$ and depth $n-1$. This was shown in [13] by considering contact terms between the currents that give rise to the charges. Hence we can write it in the form

$$\langle I_{2m-1}^n \rangle_i(\hat{\tau}) = \sum_{p=0}^{n-1} F_{2mn-2p}(\hat{\tau}) E_2(\hat{\tau})^p \,, \tag{44}$$

where $F_{2mn-2p}(\hat{\tau})$ is a weight $2mn-2p$ vector valued modular form, which can be written as a modular differential operator acting on the characters of the theory [14]. We can then take the modular transform of each term in (44) to obtain

$$\langle I_{2m-1}^n \rangle_i(\hat{\tau}) = \tau^{2mn} \langle I_{2m-1}^n \rangle_i(\tau) + \sum_{k=1}^{n-1} \left( -\frac{6i}{\pi} \right)^k \tau^{2mn-k} \sum_{p=0}^{n-k-1} F_{2mn-2(p+k)}(\tau) E_2(\tau)^p \,. \tag{45}$$

The coefficient of $\tau^{2mn-k}$ is a weight $2mn - 2k$ vector valued quasi-modular form of depth $n - k - 1$.

Take a generic correlator

$$\langle J^{(a_1)}_{2n_1-1} \ldots J^{(a_I)}_{2n_I-1} \rangle \, , \tag{46}$$

where the charges $J^{(a)}_{2n-1}$ are the zero modes on the cylinder of a weight $2n$ quasi-primary field. (We may have several quasi-primary fields of the same weight hence we have the additional index $a$. We include the KdV charges $I_{2n-1}$ in this set of charges.) This will be a weight $2 \sum_{i=1}^{I} n_i$ vector valued quasi-modular form of depth $I - 1$. Hence we expect that the $\tau^{2mn-k}$ coefficients can be written as a linear combination of correlators of the form

$$\langle J^{(a_1)}_{2n_1-1} \ldots J^{(a_{n-k})}_{2n_{n-k}-1} \rangle \, , \tag{47}$$

where $\sum_{i=1}^{n-k} n_i = mn - k$. We have see that this worked above for the case with the $I_3$ charge inserted and will see in section 4 that this works for the GGE with the $I_5$ charge in the Lee-Yang model.

Once the modular transform of each of the terms $\langle I^n_{2m-1} \rangle_i(\hat{\tau})$ has been expressed in terms of correlators of the charges $J^{(a)}_{2n-1}$ we want to re-exponentiate the expression to obtain, at least formally, an expression for the transformed GGE in terms of a new GGE. This transformed GGE will contain charges from all the quasi-primary fields in the theory, not just the KdV charges

$$\mathrm{Tr}_{\mathcal{H}_i}\left(e^{-L(I_1(R)+\alpha I_{2m-1}(R))}\right) \sim \sum_{j=1}^{N} S_{ij} \, \mathrm{Tr}_{\mathcal{H}_j}\left(\exp\left(-R\sum_{n=1}^{\infty}\sum_{a}\beta^a_{2n-1}J^{(a)}_{2n-1}(L)\right)\right), \tag{48}$$

and the $\beta^a_{2n-1}$ are constants that only depend on $\alpha$.

To end this section we note that there are two interesting cases in which we can do away with the additional charge $J_5$ appearing in (37). The first is when the charges $I_5$ and $J_5$ correspond to states which only differ by a null state (and are hence proportional to one another). This happens when $c = \frac{1}{2}$, which is the Ising Model central charge. This fact was used in the series of papers [15–17] which studied the modular properties of GGEs in the Ising model. The second case is when the central charge is $c = -2$. The integrability of the KdV equations at $c = -2$ was studied in [19], although it is not clear at the moment how one would study this is in the context of a GGE. The theory at this central charge is logarithmic, and so the GGE would involve taking traces over logarithmic modules. A review of logarithmic CFTs can be found in [20].

# 4 Asymptotic Analysis of the GGE in the Lee-Yang Model

We will now repeat the analysis from the previous section for the Lee-Yang theory. We have chosen this theory since it is arguably the simplest interacting 2d CFT with only two Virasoro representations, one with $h = 0$ and the other with $h = -1/5$. The theory therefore has two characters and they satisfy a second order modular differential equation as detailed in [21]. Using this second order differential equation allows us to simplify the expression for the correlators found in [14]. We can then use these simplified expressions to compute more of these correlators than was done in [14]. In particular we can compute to high enough order to check whether the fact that the additional charges (which come from the other quasi-primary fields) don't commute with the KdV charges stops us being able to re-exponentiate the transformed expression. In the GGE studied below with $I_5(R)$ inserted the non-commutativity is first present when we transform the $\langle I_5^6 \rangle$ term. We confirm that we can indeed still exponentiate the transformed expression to formally give an expression for the modular transforms of the original GGE as a new GGE with an infinite set of charges inserted.

In the Lee-Yang theory, the quasi-primary field that gives $I_3(R)$ is now a null state and hence the correlators containing $I_3$ vanish, as was proved in [14]. The next simplest case for a GGE here is the ensemble with $I_5(R)$ inserted

$$\text{Tr}_{\mathcal{H}_i}\left(e^{-L(\alpha I_5(R)+I_1(R))}\right). \tag{49}$$

The charges and thermal correlation functions relevant to this work have been collected in the appendices B.3 and C.2. We will just present the transformed expressions for the correlators here but all the necessary details needed to verify the results are given in B.3 and C.2.

We will proceed in the same way as the previous section and start by expanding the GGE as an asymptotic series in the chemical potential $\alpha$

$$\text{Tr}_{\mathcal{H}_i}\left(e^{-L(\alpha I_5(R)+I_1(R))}\right) = \sum_{n=0}^{\infty} \frac{1}{n!} \left(\frac{-(2\pi)^5 \alpha L}{R^5}\right)^n \langle I_5^n \rangle_i(\hat{\tau}). \tag{50}$$

Recall that $I_{2n-1} = \left(\frac{R}{2\pi}\right)^{2n-1} I_{2n-1}(R)$ and the expectation value $\langle \ldots \rangle_i$ was defined in (26). For what is to follow, we will suppress the modular $S$ matrix in our transformed expressions and we will also suppress the particular module that we are tracing over. These details are unimportant for the following discussion but can be added back in by referring to section 3.

The first few terms transform as

$$\langle 1 \rangle(\hat{\tau}) = \langle 1 \rangle(\tau), \tag{51}$$

$$\langle I_5 \rangle(\hat{\tau}) = \tau^6 \langle I_5 \rangle(\tau), \tag{52}$$

$$\langle I_5^2 \rangle(\hat{\tau}) = \tau^{12} \langle I_5^2 \rangle(\tau) - \frac{206388i}{116875\pi} \tau^{11} \langle J_9 \rangle(\tau), \tag{53}$$

$$\langle I_5^3 \rangle(\hat{\tau}) = \tau^{18} \langle I_5^3 \rangle(\tau) - \frac{619164i}{116875\pi} \tau^{17} \langle I_5 J_9 \rangle(\tau) + \tau^{16} \left(\frac{405}{4\pi^2} \langle I_{13} \rangle(\tau) + \frac{1149876}{2875\pi^2} \langle J_{13} \rangle(\tau)\right). \tag{54}$$

The charges $J_9$ and $J_{13}$ are the zero modes on the cylinder of weight 10 and 14 quasi-primary fields, respectively, that are linearly independent of the KdV charges. They are defined in terms of Virasoro modes in appendix B.3.

It is worth noting here that we did not necessarily need to use the MLDO expressions for the thermal correlators to calculate these transformations. We could have used the method developed in [22] to calculate the transformed expressions of thermal correlation functions. This method was used in, for example, [23] to calculate the transformations of $W_3$ characters in terms of zero-modes of known currents in the theory. The advantage of using the MLDO expressions comes from the fact that the map going from the currents in a 2d CFT to the thermal correlation functions of their zero-modes has a non-trivial kernel[1]. That is, if we used the method previously mentioned, then we would not know a priori whether certain parts of that expression vanished.

For example, consider the following level 9 state, which is present in any theory

$$|J_8\rangle \equiv \left(-\frac{5}{8} L_{-3}^3 + \frac{3}{2} L_{-6} L_{-3} + \frac{3}{2} L_{-4} L_{-2} L_{-3} - L_{-5} L_{-2}^2 + L_{-9} - \frac{3}{4} L_{-7} L_{-2}\right)|0\rangle. \tag{55}$$

Applying the methods outlined in appendix B, just as in the above cases, we find the associated charge to be

$$\begin{aligned} J_8(R) = \left(\frac{2\pi}{R}\right)^8 \Bigg( &\sum_{k=1}^{\infty} \left(\frac{7k^4}{4} + \frac{37k^2}{4} - \frac{59}{3}\right) L_{-k} L_k - \frac{59}{6} L_0^2 - \frac{85}{6} L_0 - \frac{1}{3} \mathcal{L}(0,0,0) \\ &-\frac{1}{6} \mathcal{L}(1,0,0) - \frac{5}{8} \mathcal{L}(1,1,1) + \frac{3}{4} \mathcal{L}(2,1,0) - \frac{1}{6} \mathcal{L}(3,0,0) \Bigg), \end{aligned} \tag{56}$$

---

[1]We would like to thank G. M. T. Watts for this observation.

where $\mathcal{L}(n, m, l)$ is defined in (222). We can verify that the thermal expectation value vanishes, and so if we had many terms appearing like this, it would be rather time-consuming to check which terms vanish in the thermal correlator and which don't. So the advantage of calculating things in terms of the MLDO is that we have non-vanishing expressions which we match to the thermal correlators of charges.

Just as before, we would like these to be the first few terms of another GGE, at least asymptotically. In essence, we would like to be able to state that the following holds asymptotically

$$\mathrm{Tr}\Big(e^{-L(\alpha I_5(R)+I_1(R))}\Big) \sim \mathrm{Tr}\Big(e^{-R(I_1(L)+\alpha_5\alpha I_5(L)+\beta_9\alpha^2 J_9(L)+\alpha_{13}\alpha^3 I_{13}(L)+\beta_{13}\alpha^3 J_{13}(L)+\dots)}\Big) , \tag{57}$$

where $\alpha_5, \beta_9, \alpha_{13}$ and $\beta_{13}$ are constants to be fixed. A priori they may depend on $\alpha, R$ and $L$, but we will see below that they are in fact numerical constants. If we write (51–54) in terms of $L$ and $R$ using $\tau = -1/\hat{\tau} = iR/L$, then comparing them with the right hand side of (57), we find

$$\alpha_5 = -1 , \tag{58}$$

$$\beta_9 = \frac{206388}{116875} , \tag{59}$$

$$\alpha_{13} = \frac{135}{2} , \tag{60}$$

$$\beta_{13} = \frac{766584}{2875} . \tag{61}$$

Given that the charges $I_{2n-1}(L)$ do not commute with the charges $J_{2n-1}(L)$, one question that may be asked is "Is this re-exponentiation a reasonable thing to do?". It seems that the answer is yes, and the fact that these charges do not commute does not affect our ability to formally re-write the transformed GGE as another GGE. Let us take some time to elaborate on this point. When we expand the right hand side of (57), the ordering of the charges in the correlators matters since they do not commute. This will lead to the presence of correlators that contain the same charges in different orders and we need to ensure that all the necessary correlators are present when we take the modular transformations of the $\langle I_5^n \rangle$ in the original GGE.

When we expand the right hand side of (57), we find that the first term that appears where the non-commutativity matters is at order $\alpha^6$ and gives us the two correlators

$$\dots + \alpha^6 \frac{R^4}{4!} \left(\frac{2\pi}{L}\right)^{28} 2\alpha_5^2 \beta_9^2 \langle I_5 J_9 I_5 J_9 + 2I_5^2 J_9^2 \rangle + \dots . \tag{62}$$

It is worth mentioning briefly that $\langle I_5 J_9 I_5 J_9 \rangle$ and $\langle I_5^2 J_9^2 \rangle$ cannot independently be written as modular linear differential operators (MLDOs) acting on the characters of the theory, but this particular linear combination presented above does have a representation as an MLDO acting on the characters. We suspect that if one carefully studies the contact terms between the relevant currents associated to these charges, as was done in [13] for a different model, then it may become clear that indeed these expectation values separately cannot be written as MLDOs, however we have not performed this analysis.

One would expect that this term appears in the transformation of the $\langle I_5^6 \rangle$ piece of the GGE as it is of weight 32 and depth 3. From Appendix C.2, we know that $\langle I_5^6 \rangle$ will be a weight 36 and depth 5 quasi-modular form that resembles

$$\langle I_5^6 \rangle = F_{36} + E_2 F_{34} + E_2^2 F_{32} + E_2^3 F_{30} + E_2^4 F_{28} + E_2^5 F_{26}, \tag{63}$$

where $F_k$ is a weight $k$ modular form. The explicit expressions for the $F_k$ in terms of differential operators acting on the characters are given in appendix C.2. After performing the modular $S$

transformation on this, we can single out the weight 32 depth 3 piece of this expression

$$\langle I_5^6 \rangle(\hat{\tau}) = \cdots - \frac{36\tau^{34}}{\pi^2}\left(10E_2^3 F_{26} + 6E_2^2 F_{28} + 3E_2 F_{30} + F_{32}\right)(\tau) + \dots . \tag{64}$$

Since this expression is a weight 32 and depth 3 quasi-modular form, we expect it to be a linear combination of the correlators

$$\langle I_5^3 J_{13}\rangle, \quad \langle I_5^3 I_{13}\rangle, \quad \langle I_5 J_9 I_5 J_9 + 2I_5^2 J_9^2\rangle, \tag{65}$$

which are all themselves weight 32 depth 3 quasi-modular forms. Using the results in appendix C.2 we find

$$10E_2^3 F_{26} + 6E_2^2 F_{28} + 3E_2 F_{30} + F_{32} = \gamma_1 \langle I_5^3 J_{13}\rangle + \gamma_2 \langle I_5^3 I_{13}\rangle + \gamma_3 \langle I_5 J_9 I_5 J_9 + 2I_5^2 J_9^2\rangle. \tag{66}$$

where

$$\gamma_1 = -\frac{127764}{575} \quad , \quad \gamma_2 = -\frac{225}{4} \quad , \quad \gamma_3 = \frac{3549667212}{2731953125}. \tag{67}$$

Therefore, in the transformed GGE we have a term of the form

$$\frac{\gamma_3}{6!}\frac{36}{\pi^2}\left(\frac{R}{L}\right)^{34}\left(-\frac{(2\pi)^5 \alpha L}{R^5}\right)^6 \langle I_5 J_9 I_5 J_9 + 2I_5^2 J_9^2\rangle. \tag{68}$$

By expanding the right hand side of (57) and comparing it with (68) we find the relation

$$\gamma_3 = \frac{5}{12}\alpha_5^2 \beta_9^2. \tag{69}$$

Using the definitions of $\gamma_3$, (67), and $\alpha_5$ and $\beta_9$, (58) and (59), we can confirm that this relation does indeed hold. Hence we have seen that at this order the fact that the charges do not commute does not prevent the re-exponentiation of the transformed expression into another (formal) GGE given by (57) and constants (58–61).

While we have found an asymptotic expression for the transformed GGE, or rather an expression with the leading charges in the transformed GGE, (57), we don't believe that these match as functions. Firstly the right hand side of (57) contains an infinite sum in the charges. It is not clear if this sum is convergent, indeed in the case of free fermions the equivalent sum over charges was not convergent and had to be regularised [15]. In the case of free fermions this regularisation introduced functions with a branch cut. Hence while the original GGE was real, the transformed expression was complex. This problem was resolved by introducing additional terms in the transformed expression that came from the thermodynamic Bethe ansatz (TBA). These additional terms made the transformed expression real. It was then proved in [17] that these additional terms gave expressions that matched exactly, not just asymptotically. We will now use the TBA for the Lee-Yang model to first reproduce our asymptotic results, and then find additional terms that we believe should be included in the transformed expression for the GGE.

# 5 Thermodynamic Bethe Ansatz for the transformed GGE

While we have found an asymptotic expression for the transformed GGE in the previous section we believe that the full expression is encoded in a set of TBA equations. We first reproduce the asymptotic results of the previous section using the TBA. We will see that when we write down the TBA equations that reproduce the asymptotics there will also be additional solutions that were missed in the asymptotic analysis. This is because these solutions give contributions

to the energy that behave as $C\alpha^{-\frac{1}{4}}$, with $\mathrm{Re}(C) > 0$, so when we exponentiate in the GGE we have terms of the form $e^{C\alpha^{-\frac{1}{4}}}$ which have a vanishing asymptotic expansion as $\alpha \to 0^-$. Hence we missed these terms in the asymptotic analysis but believe they should be included in the transformed GGE.

## 5.1 TBA and mirror TBA

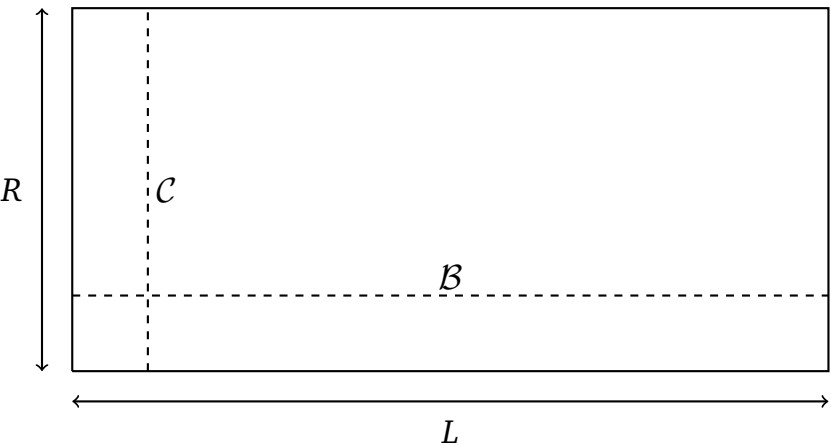

Figure 2: Strip of width $L$ and length $R$. On the horizontal slice $\mathcal{B}$ we have the Hilbert space $\mathcal{H}_\mathcal{B}$ and on the vertical slice $\mathcal{C}$ we have the Hilbert space $\mathcal{H}_\mathcal{C}$.

Let us start by considering a system living on a rectangle where the two sides have length $R$ and $L$. We will quantise our theory on the vertical slice $\mathcal{C}$, of length $R$ and treat the horizontal slice $\mathcal{B}$ as time. The partition function is then given by

$$\mathcal{Z}(R, L) = \mathrm{Tr}_{\mathcal{H}_\mathcal{C}}\left(e^{-LH_\mathcal{C}(R)}\right), \tag{70}$$

where $H_\mathcal{C}(R)$ is the Hamiltonian for the system on $\mathcal{C}$ and hence depends on $R$. For now $H_\mathcal{C}(R)$ is an arbitrary Hamiltonian but later we will take it to be either the GGE Hamiltonian or the transformed GGE Hamiltonian defined in (7) and (12). In the thermodynamic limit $L \to \infty$ we can extract the ground state energy $E_0(R)$ of $H_\mathcal{C}(R)$ via

$$\log(\mathcal{Z}(R, L)) \sim -LE_0(R), \quad L \to \infty. \tag{71}$$

If we instead quantised the system on $\mathcal{B}$ and treated $\mathcal{C}$ as the time direction then, in the thermodynamic limit, the partition function can be computed using the Bethe ansatz. This was derived in [24] and the extension to also compute the excited states was derived in [25]. We will just state the results here.

We will consider a system with only one particle species. The scattering is purely elastic and factorises into two-to-two scattering with S matrix $S(\theta)$. We will keep the form of the one particle energies $e(R, \theta)$ and momentum $p(R, \theta)$ arbitrary and we have kept the possible $R$ dependence explicit since it will be important when taking the mirror transform later.

The TBA equations for the ground state are then

$$\epsilon(\theta) = Re(R, \theta) - \int_{-\infty}^{\infty} \varphi(\theta - \theta')\log\left(1 + e^{-\epsilon(\theta')}\right)\frac{d\theta'}{2\pi}, \tag{72}$$

where $\varphi(\theta) = -i\frac{d}{d\theta}\log S(\theta)$. The ground state energy $E_0(R)$ is then given by

$$E_0(R) = -\int_{-\infty}^{\infty} \partial_\theta p(R, \theta)\log\left(1 + e^{-\epsilon(\theta)}\right)\frac{d\theta}{2\pi}. \tag{73}$$

We can also extract the excited states from the TBA equations by analytic continuation. This was first discussed in [25] and further details were given in [26]. In [25] it was conjectured that the TBA equation should be modified to

$$\epsilon(\theta) = Re(R, \theta) + \sum_{i=1}^{N} \log\left(\frac{S(\theta - \theta_i)}{S(\theta - \bar{\theta}_i)}\right) - \int_{-\infty}^{\infty} \varphi(\theta - \theta') \log\left(1 + e^{-\epsilon(\theta')}\right) \frac{d\theta'}{2\pi}, \qquad (74)$$

where the $\theta_i$ are the solutions to

$$\epsilon(\theta_i) = (2n_i + 1)\pi i, \quad n_i \in \mathbb{Z}, \qquad (75)$$

which lead to singularities in the integrand in (72). Note that there are also singularities in the integrand due to the poles in the $S$ matrix. These poles can also give rise to additional driving terms in the TBA equation (72). Solving the TBA equations with these terms added moves us between the different Virasoro representations in our theory as detailed in [25]. We will only solve TBA equations of the form (74) which gives us excited states in the ground state representation. In the Lee-Yang model this is the $h = -1/5$ representation.

When we plug the singularities $\theta_i$ into (74) we have a set of consistency conditions they must satisfy

$$2n_i\pi i = Re(R, \theta_i) - \log S(\theta_i - \bar{\theta}_i) + \sum_{\substack{j=1 \\ j\neq i}}^{N} \log\left(\frac{S(\theta_i - \theta_j)}{S(\theta_i - \bar{\theta}_j)}\right) - \int_{-\infty}^{\infty} \varphi(\theta_i - \theta') \log\left(1 + e^{-\epsilon(\theta')}\right) \frac{d\theta'}{2\pi}. \qquad (76)$$

The specific choice of branch cuts of the logarithms won't matter in our analysis but they have been carefully studied in [26]. The excited state energy is then given by

$$E(R) = i \sum_{i=1}^{N} (p(R, \bar{\theta}_i) - p(R, \theta_i)) - \int_{-\infty}^{\infty} \partial_\theta p(R, \theta) \log\left(1 + e^{-\epsilon(\theta)}\right) \frac{d\theta}{2\pi}. \qquad (77)$$

When we numerically solve the TBA equations for the Lee-Yang model we will only do it for the ground state and excited states corresponding to $N = 1$.

We are interested in the modular transform

$$S : \hat{\tau} = \frac{iL}{R} \mapsto \frac{iR}{L} = \tau, \qquad (78)$$

which swaps the cycles $\mathcal{C}$ and $\mathcal{B}$ in figure 2. Since we have swapped $\mathcal{C}$ and $\mathcal{B}$ we are now interested in the spectrum of the Hamiltonian $H_{\mathcal{B}}(L)$ which acts on the Hilbert space $\mathcal{H}_{\mathcal{B}}$. The spectrum can again be found by solving TBA equations. The energy and momentum of the new system is given by the mirror transform of the original TBA. The mirror energy and momentum are denoted by $\widetilde{e}(L, \theta)$ and $\widetilde{p}(L, \theta)$ respectively and are related to the original energy and momentum by

$$\widetilde{e}(L, \theta) = ip\left(L, \theta - \frac{i\pi}{2}\right), \quad \widetilde{p}(L, \theta) = ie\left(L, \theta - \frac{i\pi}{2}\right) \qquad (79)$$

The TBA equations for the ground state, $\widetilde{E}_0(L)$, of the modular transformed theory are

$$\widetilde{\epsilon}(\theta) = L\widetilde{e}(L, \theta) - \int_{-\infty}^{\infty} \varphi(\theta - \theta') \log\left(1 + e^{-\widetilde{\epsilon}(\theta')}\right) \frac{d\theta'}{2\pi}, \qquad (80)$$

$$\widetilde{E}_0(L) = -\int_{-\infty}^{\infty} \partial_\theta \widetilde{p}(L, \theta) \log\left(1 + e^{-\widetilde{\epsilon}(\theta)}\right) \frac{d\theta}{2\pi}. \qquad (81)$$

469 and the excited states are given by

$$\widetilde{\epsilon}(\theta) = L\widetilde{e}(L,\theta) + \sum_{i=1}^{N} \log\left(\frac{S(\theta-\theta_i)}{S(\theta-\bar{\theta}_i)}\right) - \int_{-\infty}^{\infty} \varphi(\theta-\theta')\log\left(1+e^{-\widetilde{\epsilon}(\theta')}\right)\frac{d\theta'}{2\pi}, \qquad (82)$$

$$\widetilde{E}(L) = i\sum_{i=1}^{N}(\widetilde{p}(L,\bar{\theta}_i)-\widetilde{p}(L,\theta_i)) - \int_{-\infty}^{\infty} \partial_\theta\widetilde{p}(L,\theta)\log\left(1+e^{-\widetilde{\epsilon}(\theta')}\right)\frac{d\theta}{2\pi}. \qquad (83)$$

470 We again have a constraint equation that the $\theta_i$ must satisfy

$$2n_i\pi i = L\widetilde{e}(L,\theta_i) - \log S(\theta_i-\bar{\theta}_i) + \sum_{\substack{j=1\\j\neq i}}^{N} \log\left(\frac{S(\theta_i-\theta_j)}{S(\theta_i-\bar{\theta}_j)}\right) - \int_{-\infty}^{\infty} \varphi(\theta_i-\theta')\log\left(1+e^{-\widetilde{\epsilon}(\theta')}\right)\frac{d\theta'}{2\pi}, \tag{84}$$

471 where $n_i \in \mathbb{Z}$.

## 5.2  TBA for the GGE

473 First we will use the TBA equations to reproduce the spectrum of the GGE with the $I_5(R)$ charge
474 inserted. The definition of the spectrum of the GGE was given in (9). The S matrix $S(\theta)$ for
475 the Lee-Yang model is

$$S(\theta) = \frac{\sinh(\theta)+i\sin(\frac{\pi}{3})}{\sinh(\theta)-i\sin(\frac{\pi}{3})}. \qquad (85)$$

476 To reproduce the spectrum we set the one particle energy $e(R,\theta)$ and momentum $p(R,\theta)$ to
477 be

$$e(R,\theta) = \frac{1}{R}e^\theta, \quad p(R,\theta) = \frac{1}{R}e^\theta + \frac{\alpha C}{R^5}e^{5\theta}. \qquad (86)$$

478 where the constant $C$ is

$$C = -\frac{32400\sqrt{3}\pi^2\Gamma(\frac{2}{3})^6}{1729\Gamma(\frac{1}{6})^6}. \qquad (87)$$

479 The constant $C$ can be computed using the results in [18], in particular

$$C = -\left(\frac{2\pi}{R}\right)^5\frac{4}{5C_3\kappa^5}\sin\left(\frac{8\pi}{3}\right), \qquad (88)$$

480 where $C_3$ is given in equation (4.35), $\kappa$ in (4.16) and the combination is given in (4.34) in [18].
481 (Note that our TBA equation (89) differs from (4.30) in [18] where the driving term is $\kappa e^\theta$
482 instead of $e^\theta$. This accounts for the factor $\kappa^5$ in (88).)
483     The TBA equation for the ground state is

$$\epsilon(\theta) = e^\theta - \int_{-\infty}^{\infty} \varphi(\theta-\theta')\log\left(1+e^{-\epsilon(\theta')}\right)\frac{d\theta'}{2\pi}, \qquad (89)$$

484 and the ground state energy is given by

$$E_0(R) = -\int_{-\infty}^{\infty}\left(\frac{1}{R}e^\theta + \frac{5\alpha C}{R^5}e^{5\theta}\right)\log\left(1+e^{-\epsilon(\theta)}\right)\frac{d\theta}{2\pi}. \qquad (90)$$

485 The $\alpha^0$ term in the integral gives the vacuum eigenvalue of $I_1(R)$ in the $h=-\frac{1}{5}$ representation
486 and the $\alpha$ term gives the vacuum eigenvalue of $I_5(R)$ for $h=-\frac{1}{5}$. This was derived in [18].
487 We have started with the TBA equations for a massless theory, however we could start with a

massive theory and then take the massless limit. This was done in [27] and gives the same TBA equations we are studying here.

The excited states TBA equations are

$$\epsilon(\theta) = e^{\theta} + \sum_{i=1}^{N} \log\left(\frac{S(\theta - \theta_i)}{S(\theta - \bar{\theta}_i)}\right) - \int_{-\infty}^{\infty} \varphi(\theta - \theta') \log\left(1 + e^{-\epsilon(\theta')}\right) \frac{d\theta'}{2\pi} \,, \qquad (91)$$

and the energies are given by the integrals

$$E(R) = i \sum_{i=1}^{N} \left(\frac{1}{R}\left(e^{\bar{\theta}_i} - e^{\theta_i}\right) + \frac{\alpha C}{R^5}\left(e^{5\bar{\theta}_i} - e^{5\theta_i}\right)\right) - \int_{-\infty}^{\infty} \left(\frac{1}{R}e^{\theta} + \frac{5\alpha C}{R^5}e^{5\theta}\right) \log\left(1 + e^{-\epsilon(\theta)}\right) \frac{d\theta}{2\pi} \,. \tag{92}$$

The $\theta_i$ satisfy the constraints

$$2 n_i \pi i = e^{\theta_i} - \log S(\theta_i - \bar{\theta}_i) + \sum_{\substack{j=1 \\ j \neq i}}^{N} \log\left(\frac{S(\theta_i - \theta_j)}{S(\theta_i - \bar{\theta}_j)}\right) - \int_{-\infty}^{\infty} \varphi(\theta_i - \theta') \log\left(1 + e^{-\epsilon(\theta')}\right) \frac{d\theta'}{2\pi} \,. \tag{93}$$

It was again verified in [18] that solving these TBA equations gives the excited state eigenvalues for $I_1(R)$ and $I_5(R)$.

## 5.3   Transformed TBA

We now want to find the spectrum of the modular transformed GGE, which was defined in (14). As discussed above in section 5.1, if we know the TBA equations that encode the spectrum of the GGE then to find the spectrum of the transformed GGE we use the mirror TBA. The mirror energy $\widetilde{e}(L, \theta)$ and momentum $\widetilde{p}(L, \theta)$ were given in (79). Using the explicit forms of the energy and momentum for the original GGE (86), the mirror energy and momentum are

$$\widetilde{e}(L, \theta) = \frac{1}{L}e^{\theta} + \frac{\alpha C}{L^5}e^{5\theta} \,, \quad \widetilde{p}(L, \theta) = \frac{1}{L}e^{\theta} \tag{94}$$

Hence the TBA equation for the ground state is

$$\epsilon(\theta) = e^{\theta} + \frac{\alpha C}{L^4}e^{5\theta} - \int_{-\infty}^{\infty} \varphi(\theta - \theta') \log\left(1 + e^{-\epsilon(\theta')}\right) \frac{d\theta'}{2\pi} \,, \tag{95}$$

and the ground state energy is given by

$$E_0(L) = -\frac{1}{L} \int_{-\infty}^{\infty} e^{\theta} \log\left(1 + e^{-\epsilon(\theta)}\right) \frac{d\theta}{2\pi} \,. \tag{96}$$

(Note we have dropped the tilde from $\epsilon$ which we had in (80) and (81) to distinguish the mirror TBA from the original TBA equations.) The excited state mirror TBA equations are

$$\epsilon(\theta) = e^{\theta} + \frac{\alpha C}{L^4}e^{5\theta} + \sum_{i=1}^{N} \log\left(\frac{S(\theta - \theta_i)}{S(\theta - \bar{\theta}_i)}\right) - \int_{-\infty}^{\infty} \varphi(\theta - \theta') \log\left(1 + e^{-\epsilon(\theta')}\right) \frac{d\theta'}{2\pi} \,, \tag{97}$$

and the energies are given by the integrals

$$E(L) = \frac{i}{L} \sum_{i=1}^{N} \left(e^{\bar{\theta}_i} - e^{\theta_i}\right) - \frac{1}{L} \int_{-\infty}^{\infty} e^{\theta} \log\left(1 + e^{-\epsilon(\theta)}\right) \frac{d\theta}{2\pi} \,. \tag{98}$$

Again the $\theta_i$ satisfy the constraints

$$2n_i\pi i = e^{\theta_i} + \frac{\alpha C}{L^4}e^{5\theta_i} - \log S(\theta_i - \bar{\theta}_i) + \sum_{\substack{j=1 \\ j \neq i}}^{N} \log\left(\frac{S(\theta_i - \theta_j)}{S(\theta_i - \bar{\theta}_j)}\right) - \int_{-\infty}^{\infty} \varphi(\theta_i - \theta') \log\left(1 + e^{-\epsilon(\theta')}\right)\frac{d\theta'}{2\pi}.$$
(99)

The constant $C$ define in (87) is negative. Hence we only have solutions to the TBA equations (95) and (97) if $\text{Re}(\alpha) < 0$. Otherwise the $\log\left(1 + e^{-\epsilon(\theta')}\right)$ term in the convolution integrals will diverge, since for large $\theta > 0$ it behaves as $\log\left(1 + e^{-\alpha C e^{5\theta}/L^4}\right)$. Throughout the following sections we will only consider $\alpha$ on the negative real axis.

We will numerically check in the next section that these TBA equations for both the ground state and the excited states reproduce the spectrum of the transformed GGE found from the asymptotic analysis. We will then show how to find other solutions to the TBA equations which do not appear in the asymptotic analysis. We will conjecture that these are all the solutions and including all of them reproduces the full spectrum of the transformed GGE.

## 5.4 Asymptotic results from the TBA

We want to show that the TBA equations (95), (96) and (97), (98) reproduce the asymptotic spectrum found in section 4. From (57–61) we expect the ground state energy $E_0(L)$ in the transformed GGE to have the asymptotic expansion

$$E_0(L) \sim \mathcal{I}_1^{\text{vac}}(L) - \alpha\mathcal{I}_5^{\text{vac}}(L) + \beta_9\alpha^2\mathcal{J}_9^{\text{vac}}(L) + \alpha^3\left(\alpha_{13}\mathcal{I}_{13}^{\text{vac}}(L) + \beta_{13}\mathcal{J}_{13}^{\text{vac}}(L)\right) + O(\alpha^4), \quad (100)$$

where the $\alpha_{2n-1}$ and $\beta_{2n-1}$ are given in (59–61) and $\mathcal{I}_{2n-1}^{\text{vac}}(L)$ is the eigenvalue of the charge $I_{2n-1}(L)$ on the highest weight state $|-1/5\rangle$ and similarly for $\mathcal{J}_{2n-1}^{\text{vac}}(L)$.

In order to reproduce this asymptotic expansion for $E_0(L)$ defined in (96), we assume that the pseudo energy $\epsilon(\theta)$ has the asymptotic expansion

$$\epsilon(\theta) \sim \sum_{n=0}^{\infty} \epsilon_n(\theta)\left(\frac{\alpha}{L^4}\right)^n. \quad (101)$$

Recall that we mentioned in the previous section that the TBA equations only have solutions for $\text{Re}(\alpha) < 0$ hence the expansion (101) must have zero radius of convergence and is therefore asymptotic. We also define the function

$$L(\epsilon(\theta)) = \log\left(1 + e^{-\epsilon(\theta)}\right). \quad (102)$$

Plugging the asymptotic expansion for $\epsilon$ in $L(\epsilon)$ gives

$$
\begin{aligned}
L(\epsilon) \sim\ & L(\epsilon_0) + \frac{\alpha}{L^4}\epsilon_1 L'(\epsilon_0) + \frac{\alpha^2}{L^8}\left(\epsilon_2 L'(\epsilon_0) + \frac{1}{2}\epsilon_1^2 L''(\epsilon_0)\right) \\
& + \frac{\alpha^3}{L^{12}}\left(\epsilon_3 L'(\epsilon_0) + \epsilon_2\epsilon_1 L''(\epsilon_0) + \frac{1}{6}\epsilon_1^3 L'''(\epsilon_0)\right) + O(\alpha^4).
\end{aligned}
$$
(103)

If we then use these asymptotic expansions in the TBA equation (95) and collect each power

of $\alpha$ we end up with the series of equations

$$\epsilon_0(\theta) = e^\theta - \int_{-\infty}^{\infty} \varphi(\theta - \theta') L(\epsilon_0(\theta')) \frac{d\theta'}{2\pi} , \tag{104}$$

$$\epsilon_1(\theta) = Ce^{5\theta} - \int_{-\infty}^{\infty} \varphi(\theta - \theta') \epsilon_1(\theta') L'(\epsilon_0(\theta')) \frac{d\theta'}{2\pi} , \tag{105}$$

$$\epsilon_2(\theta) = - \int_{-\infty}^{\infty} \varphi(\theta - \theta') \left( \epsilon_2(\theta') L'(\epsilon_0(\theta')) + \frac{1}{2} \epsilon_1(\theta')^2 L''(\epsilon_0(\theta')) \right) \frac{d\theta'}{2\pi} , \tag{106}$$

$$\epsilon_3(\theta) = - \int_{-\infty}^{\infty} \varphi(\theta - \theta') \left( \epsilon_3(\theta') L'(\epsilon_0(\theta')) + \epsilon_2(\theta') \epsilon_1(\theta') L''(\epsilon_0(\theta')) + \frac{1}{6} \epsilon_1(\theta')^3 L'''(\epsilon_0(\theta')) \right) \frac{d\theta'}{2\pi}, \tag{107}$$

Note that the first equation (104) is the usual TBA equation for a massless theory. Once we have solved (104) we can then treat $\epsilon_0$ as a known function in (105). Hence (105) is a linear equation in $\epsilon_1$. We can continue to iteratively solve the TBA equations for $\epsilon_n$ with $n \geq 2$. For $n \geq 1$ the TBA equations take the general form

$$\epsilon_n(\theta) = f_n(\theta) - \int_{-\infty}^{\infty} \varphi(\theta - \theta') \epsilon_n(\theta') L'(\epsilon_0(\theta')) \frac{d\theta'}{2\pi} , \tag{108}$$

where the functions $f_n(\theta)$ depend on $\epsilon_k(\theta)$ for $k = 0, \ldots, n-1$ which have been previously solved for. These are again linear integral equations for $\epsilon_n(\theta)$. We will outline how to solve these equations numerically in appendix D.

We can similarly expand the ground state energy (96) in $\alpha$ to obtain the asymptotic expansion

$$\begin{aligned}
E_0(L) = &-\frac{1}{L} \int_{-\infty}^{\infty} e^\theta L(\epsilon_0(\theta)) \frac{d\theta}{2\pi} - \frac{\alpha}{L^5} \int_{-\infty}^{\infty} e^\theta \epsilon_1(\theta) L'(\epsilon_0(\theta)) \frac{d\theta}{2\pi} \\
&- \frac{\alpha^2}{L^9} \int_{-\infty}^{\infty} e^\theta \left( \epsilon_2(\theta) L'(\epsilon_0(\theta)) + \frac{1}{2} \epsilon_1(\theta)^2 L''(\epsilon_0(\theta)) \right) \frac{d\theta}{2\pi} \\
&- \frac{\alpha^3}{L^{13}} \int_{-\infty}^{\infty} e^\theta \left( \epsilon_3(\theta) L'(\epsilon_0(\theta)) + \epsilon_2(\theta) \epsilon_1(\theta) L''(\epsilon_0(\theta)) + \frac{1}{6} \epsilon_1(\theta)^3 L'''(\epsilon_0(\theta)) \right) \frac{d\theta}{2\pi} \\
&+ O(\alpha^4) .
\end{aligned} \tag{109}$$

If we compare this with (100) we find that the following relations must hold

$$\mathcal{I}_1^{\text{vac}}(L) = -\frac{1}{L} \int_{-\infty}^{\infty} e^\theta L(\epsilon_0(\theta)) \frac{d\theta}{2\pi} , \tag{110}$$

$$\mathcal{I}_5^{\text{vac}}(L) = \frac{1}{L^5} \int_{-\infty}^{\infty} e^\theta \epsilon_1(\theta) L'(\epsilon_0(\theta)) \frac{d\theta}{2\pi} , \tag{111}$$

$$\beta_9 \mathcal{J}_9^{\text{vac}}(L) = -\frac{1}{L^9} \int_{-\infty}^{\infty} e^\theta \left( \epsilon_2(\theta) L'(\epsilon_0(\theta)) + \frac{1}{2} \epsilon_1(\theta)^2 L''(\epsilon_0(\theta)) \right) \frac{d\theta}{2\pi} , \tag{112}$$

$$\alpha_{13} \mathcal{I}_{13}^{\text{vac}}(L) + \beta_{13} \mathcal{J}_{13}^{\text{vac}}(L) = \tag{113}$$

$$-\frac{1}{L^{13}} \int_{-\infty}^{\infty} e^\theta \left( \epsilon_3(\theta) L'(\epsilon_0(\theta)) + \epsilon_2(\theta) \epsilon_1(\theta) L''(\epsilon_0(\theta)) + \frac{1}{6} \epsilon_1^3(\theta) L'''(\epsilon_0(\theta)) \right) \frac{d\theta}{2\pi} , \tag{114}$$

where the numerical constants $\alpha_{2n-1}$ and $\beta_{2n-1}$ are given in (59–61). We note from (104–107) that the pseudo energies are independent of $L$ and hence we have the correct $L$ dependence in for the charges.

543    We have numerically solved the TBA equations for the ground state and collected the results
544  in section 5.5.

545    The results in section 4 also give an asymptotic expansion for the excited states in the
546  transformed GGE. The excited states are given by the TBA equations (97) and (98) along with
547  the constraint (99). We will focus on the case where we have picked up just one pole in the
548  equations, which we will denote by $\eta$. Then the TBA equation (97) becomes

$$\epsilon(\theta) = e^{\theta} + \frac{\alpha C}{L^4} e^{5\theta} + \log\left(\frac{S(\theta-\eta)}{S(\theta-\bar{\eta})}\right) - \int_{-\infty}^{\infty} \varphi(\theta-\theta')\log\left(1+e^{-\epsilon(\theta')}\right)\frac{d\theta'}{2\pi}, \qquad (115)$$

549  the energy (98) becomes

$$E(L) = \frac{i}{L}\left(e^{\bar{\eta}} - e^{\eta}\right) - \frac{1}{L}\int_{-\infty}^{\infty} e^{\theta}\log\left(1+e^{-\epsilon(\theta)}\right)\frac{d\theta}{2\pi}, \qquad (116)$$

550  and the constraint (99) becomes

$$2n\pi i = e^{\eta} + \frac{\alpha C}{L^4}e^{5\eta} - \log S(2i\mathrm{Im}(\eta)) - \int_{-\infty}^{\infty} \varphi(\eta-\theta')\log\left(1+e^{-\epsilon(\theta')}\right)\frac{d\theta'}{2\pi}. \qquad (117)$$

551  To find an asymptotic solution we will again assume that $\epsilon$ has the asymptotic expansion (101).
552  Furthermore, we will assume that the pole $\eta$ also has an asymptotic expansion

$$\eta \sim \sum_{n=0}^{\infty} \eta_n \left(\frac{\alpha}{L^4}\right)^n. \qquad (118)$$

553  Using (118) we have the following asymptotic expansions. First we expand $\log\left(\frac{S(\theta-\eta)}{S(\theta-\bar{\eta})}\right)$ which
554  appears in (115). We note $\log\left(\frac{S(\theta-\eta)}{S(\theta-\bar{\eta})}\right) = 2\mathrm{Re}\left(\log S(\theta-\eta)\right)$ for $\theta \in \mathbb{R}$ and hence

$$\begin{aligned}
\log\left(\frac{S(\theta-\eta)}{S(\theta-\bar{\eta})}\right) = &\, 2\mathrm{Re}\left(\log S(\theta-\eta_0)\right) + 2\frac{\alpha}{L^4}\mathrm{Im}(\eta_1\varphi(\theta-\eta_0)) \\
&+ 2\frac{\alpha^2}{L^8}\mathrm{Im}\left(\eta_2\varphi(\theta-\eta_0) - \frac{1}{2}\eta_1^2\varphi'(\theta-\eta_0)\right) \\
&+ 2\frac{\alpha^3}{L^{12}}\mathrm{Im}\left(\eta_3\varphi(\theta-\eta_0) - \eta_2\eta_1\varphi'(\theta-\eta_0) + \frac{1}{6}\eta_1^3\varphi''(\theta-\eta_0)\right) + O(\alpha^4)
\end{aligned} \qquad (119)$$

555  We also need the expansions of $\log S(2i\mathrm{Im}(\eta))$ and $\varphi(\eta-\theta')$ in (117)

$$\log S(2i\mathrm{Im}(\eta)) = \qquad (120)$$

$$\log S(2i\mathrm{Im}(\eta_0)) - 2\frac{\alpha}{L^4}\mathrm{Im}(\eta_1)\varphi(2i\mathrm{Im}(\eta_0)) - 2\frac{\alpha^2}{L^8}\left(\mathrm{Im}(\eta_2)\varphi(2i\mathrm{Im}(\eta_0)) + i\mathrm{Im}(\eta_1)^2\varphi'(2i\mathrm{Im}(\eta_0))\right)$$

$$- \frac{\alpha^3}{L^{12}}\left(2\mathrm{Im}(\eta_3)\varphi(2i\mathrm{Im}(\eta_0)) + 4i\mathrm{Im}(\eta_2)\mathrm{Im}(\eta_1)\varphi'(2i\mathrm{Im}(\eta_0)) - \frac{4}{3}\mathrm{Im}(\eta_1)^3\varphi''(2i\mathrm{Im}(\eta_0))\right) + O(\alpha^4),$$

556  and

$$\begin{aligned}
\varphi(\eta-\theta') = &\, \varphi(\eta_0-\theta') + \frac{\alpha}{L^4}\eta_1\varphi'(\eta_0-\theta') + \frac{\alpha^2}{L^8}\left(\eta_2\varphi'(\eta_0-\theta') + \frac{1}{2}\eta_1^2\varphi''(\eta_0-\theta')\right) \\
&+ \frac{\alpha^3}{L^{12}}\left(\eta_3\varphi'(\eta_0-\theta') + \eta_2\eta_1\varphi''(\eta_0-\theta') + \frac{1}{6}\eta_1^3\varphi'''(\eta_0-\theta')\right) + O(\alpha^4). \quad (121)
\end{aligned}$$

Finally we also expand the exponentials $e^\eta$ and $e^{5\eta}$. Plugging these expansions into (115) gives us the series of equations

$$\epsilon_0(\theta) = e^\theta + \log\left(\frac{S(\theta - \eta_0)}{S(\theta - \bar{\eta}_0)}\right) - \int_{-\infty}^{\infty} \varphi(\theta - \theta')\log\left(1 + e^{-\epsilon_0(\theta')}\right)\frac{d\theta'}{2\pi}, \tag{122}$$

$$\epsilon_1(\theta) = Ce^{5\theta} + 2\text{Im}(\eta_1\varphi(\theta - \eta_0)) - \int_{-\infty}^{\infty} \varphi(\theta - \theta')\epsilon_1(\theta')L'(\epsilon_0(\theta'))\frac{d\theta'}{2\pi}, \tag{123}$$

$$\epsilon_2(\theta) = 2\text{Im}\left(\eta_2\varphi(\theta - \eta_0) - \frac{1}{2}\eta_1^2\varphi'(\theta - \eta_0)\right) \tag{124}$$

$$- \int_{-\infty}^{\infty} \varphi(\theta - \theta')\left(\epsilon_2(\theta')L'(\epsilon_0(\theta')) + \frac{1}{2}\epsilon_1(\theta')^2L''(\epsilon_0(\theta'))\right)\frac{d\theta'}{2\pi},$$

$$\epsilon_3(\theta) = 2\text{Im}\left(\eta_3\varphi(\theta - \eta_0) - \eta_2\eta_1\varphi'(\theta - \eta_0) + \frac{1}{6}\eta_1^3\varphi''(\theta - \eta_0)\right) \tag{125}$$

$$- \int_{-\infty}^{\infty} \varphi(\theta - \theta')\left(\epsilon_3(\theta')L'(\epsilon_0(\theta')) + \epsilon_2(\theta')\epsilon_1(\theta')L''(\epsilon_0(\theta')) + \frac{1}{6}\epsilon_1^3(\theta')L'''(\epsilon_0(\theta'))\right)\frac{d\theta'}{2\pi}.$$

For $n \geq 1$ the TBA equations for $\epsilon_n$ take the form

$$\epsilon_n(\theta) = g_n(\eta_0, \ldots, \eta_{n-1}; \epsilon_0, \ldots, \epsilon_{n-1}; \theta) + 2\text{Im}(\eta_n\varphi(\theta - \eta_0)) - \int_{-\infty}^{\infty} \varphi(\theta - \theta')\epsilon_n(\theta')L'(\epsilon_0(\theta'))\frac{d\theta'}{2\pi}, \tag{126}$$

where $g_n$ contains all the dependence on the previously determined $\eta_i$ and $\epsilon_i$. We will use this when we discuss how to numerically solve the TBA equations in appendix D.

We can similarly plug the expansions into the constraint equation (117) and obtain the system of constraints

$$2n\pi i = e^{\eta_0} - \log S(2i\text{Im}(\eta_0)) - \int_{-\infty}^{\infty} \varphi(\eta_0 - \theta')\log\left(1 + e^{-\epsilon_0(\theta')}\right)\frac{d\theta'}{2\pi}, \tag{127}$$

$$0 = \eta_1 e^{\eta_0} + Ce^{5\eta_0} + 2\text{Im}(\eta_1)\varphi(2i\text{Im}(\eta_0))$$

$$- \int_{-\infty}^{\infty} \left(\eta_1\varphi'(\eta_0 - \theta')L(\epsilon_0(\theta')) + \varphi(\eta_0 - \theta')\epsilon_1(\theta')L'(\epsilon_0(\theta'))\right)\frac{d\theta'}{2\pi}, \tag{128}$$

$$0 = \left(\eta_2 + \frac{1}{2}\eta_1^2\right)e^{\eta_0} + 5C\eta_1 e^{5\eta_0} + 2\left(\text{Im}(\eta_2)\varphi(2i\text{Im}(\eta_0)) + i\text{Im}(\eta_1)^2\varphi'(2i\text{Im}(\eta_0))\right)$$

$$- \int_{-\infty}^{\infty} \left(\left(\eta_2\varphi'(\eta_0 - \theta') + \frac{1}{2}\eta_1^2\varphi''(\eta_0 - \theta')\right)L(\epsilon_0(\theta')) + \eta_1\varphi'(\eta_0 - \theta')\epsilon_1(\theta')L'(\epsilon_0(\theta'))\right.$$

$$\left. + \varphi(\eta_0 - \theta')\left(\epsilon_2(\theta')L'(\epsilon_0(\theta')) + \frac{1}{2}\epsilon_1(\theta')^2L''(\epsilon_0(\theta'))\right)\right)\frac{d\theta'}{2\pi}, \tag{129}$$

566
$$0 = \left(\eta_3 + \eta_2\eta_1 + \frac{1}{6}\eta_1^3\right)e^{\eta_0} + C\left(5\eta_2 + \frac{25}{2}\eta_1^2\right)e^{5\eta_0}$$
$$+ \left(2\text{Im}(\eta_3)\varphi(2i\text{Im}(\eta_0)) + 4i\text{Im}(\eta_2)\text{Im}(\eta_1)\varphi'(2i\text{Im}(\eta_0)) - \frac{4}{3}\text{Im}(\eta_1)^3\varphi''(2i\text{Im}(\eta_0))\right)$$
$$- \int_{-\infty}^{\infty}\left(\left(\eta_3\varphi'(\eta_0 - \theta') + \eta_2\eta_1\varphi''(\eta_0 - \theta') + \frac{1}{6}\eta_1^3\varphi'''(\eta_0 - \theta')\right)L(\epsilon_0(\theta'))\right.$$
$$+ \left(\eta_2\varphi'(\eta_0 - \theta') + \frac{1}{2}\eta_1^2\varphi''(\eta_0 - \theta')\right)\epsilon_1(\theta')L'(\epsilon_0(\theta'))$$
$$+ \eta_1\varphi'(\eta_0 - \theta')\left(\epsilon_2(\theta')L'(\epsilon_0(\theta')) + \frac{1}{2}\epsilon_1(\theta')^2L''(\epsilon_0(\theta'))\right)$$
$$+ \varphi(\eta_0 - \theta')\left(\epsilon_3(\theta')L'(\epsilon_0(\theta')) + \epsilon_2(\theta')\epsilon_1(\theta')L''(\epsilon_0(\theta')) + \frac{1}{6}\epsilon_1^3(\theta')L'''(\epsilon_0(\theta'))\right)\right)\frac{d\theta'}{2\pi}.$$

567 For $n \geq 1$, the constraint equation determining $\eta_n$ and $\epsilon_n$ is given by

$$0 = h_n(\eta_0, \ldots, \eta_{n-1}; \epsilon_0, \ldots, \epsilon_{n-1}) + \eta_n e^{\eta_0} + 2\text{Im}(\eta_n)\varphi(2i\text{Im}(\eta_0)) \qquad (130)$$
$$- \int_{-\infty}^{\infty}\left(\eta_n\varphi'(\eta_0 - \theta)L(\epsilon_0(\theta)) + \epsilon_n(\theta)\varphi(\eta_0 - \theta)L'(\epsilon_0(\theta))\right)\frac{d\theta}{2\pi},$$

568 where $h_n$ contains all the dependence on the previously determined $\eta_i$ and $\epsilon_i$. We will explain
569 how to numerically solve the constraint equation in appendix D.
570  Finally if we expand the energy (116) we obtain the asymptotic expansion

$$E(L) \sim \frac{1}{L}\left(2\text{Im}(e^{\eta_0}) - \int_{-\infty}^{\infty}e^{\theta}L(\epsilon_0(\theta))\frac{d\theta}{2\pi}\right) \qquad (131)$$
$$+ \frac{\alpha}{L^5}\left(2\text{Im}(\eta_1 e^{\eta_0}) - \int_{-\infty}^{\infty}e^{\theta}\epsilon_1(\theta)L'(\epsilon_0(\theta))\frac{d\theta}{2\pi}\right)$$
$$+ \frac{\alpha^2}{L^9}\left(2\text{Im}\left(\left(\eta_2 + \frac{1}{2}\eta_1^2\right)e^{\eta_0}\right) - \int_{-\infty}^{\infty}e^{\theta}\left(\epsilon_2(\theta)L'(\epsilon_0(\theta)) + \frac{1}{2}\epsilon_1(\theta)^2L''(\epsilon_0(\theta))\right)\frac{d\theta}{2\pi}\right)$$
$$+ \frac{\alpha^3}{L^{13}}\left(2\text{Im}\left(\left(\eta_3 + \eta_2\eta_1 + \frac{1}{6}\eta_1^3\right)e^{\eta_0}\right)\right.$$
$$\left.- \int_{-\infty}^{\infty}e^{\theta}\left(\epsilon_3(\theta)L'(\epsilon_0(\theta)) + \epsilon_2(\theta)\epsilon_1(\theta)L''(\epsilon_0(\theta)) + \frac{1}{6}\epsilon_1^3(\theta)L'''(\epsilon_0(\theta))\right)\frac{d\theta}{2\pi}\right) + O(\alpha^4)$$

571 For levels 1,2 and 3 in the $h = -1/5$ representation of the Lee-Yang model we have only one
572 state. Hence using (57–61) we see that the coefficients in the expansion are related to the
573 single eigenvalue of the charges $I_{2n-1}$ and $J_{2n-1}$ as follows

$$\mathcal{I}_1(L) = \frac{1}{L}\left(2\text{Im}(e^{\eta_0}) - \int_{-\infty}^{\infty}e^{\theta}L(\epsilon_0(\theta))\frac{d\theta}{2\pi}\right), \qquad (132)$$

$$\mathcal{I}_5(L) = -\frac{1}{L^5}\left(2\text{Im}(\eta_1 e^{\eta_0}) - \int_{-\infty}^{\infty}e^{\theta}\epsilon_1(\theta)L'(\epsilon_0(\theta))\frac{d\theta}{2\pi}\right), \qquad (133)$$

$$\beta_9\mathcal{J}_9(L) = \frac{1}{L^9}\left(2\text{Im}\left(\left(\eta_2 + \frac{1}{2}\eta_1^2\right)e^{\eta_0}\right) - \int_{-\infty}^{\infty}e^{\theta}\left(\epsilon_2(\theta)L'(\epsilon_0(\theta)) + \frac{1}{2}\epsilon_1(\theta)^2L''(\epsilon_0(\theta))\right)\frac{d\theta}{2\pi}\right), \qquad (134)$$

$$\alpha_{13}\mathcal{I}_{13}(L) + \beta_{13}\mathcal{J}_{13}(L) = \frac{1}{L^{13}}\left(2\text{Im}\left(\left(\eta_3 + \eta_2\eta_1 + \frac{1}{6}\eta_1^3\right)e^{\eta_0}\right)\right.$$
$$\left.- \int_{-\infty}^{\infty}e^{\theta}\left(\epsilon_3(\theta)L'(\epsilon_0(\theta)) + \epsilon_2(\theta)\epsilon_1(\theta)L''(\epsilon_0(\theta)) + \frac{1}{6}\epsilon_1^3(\theta)L'''(\epsilon_0(\theta))\right)\frac{d\theta}{2\pi}\right). \qquad (135)$$

574 Here the $\mathcal{I}_{2n-1}(L)$ and $\mathcal{J}_{2n-1}(L)$ are eigenvalues of the charges $I_{2n-1}(L)$ and $J_{2n-1}(L)$ in the
575 excited states. Again these relations only apply to the case where we have a single state at a
576 given level in the Virasoro representation.

577      For level 4 and higher we have multiple states in the $h = -1/5$ representation and we
578 need to be more careful. The coefficients in the expansion (131) of the energy $E(L)$ will no
579 longer be given by eigenvalues of the individual charges since the charges don't commute and
580 therefore can't be simultaneously diagonalised.

581      Recall that we want to reproduce the right hand side of (57)

$$\text{Tr}\Big(e^{-R(I_1(L)-\alpha I_5(L)+\beta_9\alpha^2 J_9(L)+\alpha_{13}\alpha^3 I_{13}(L)+\beta_{13}\alpha^3 J_{13}(L)+\dots)}\Big)\,, \tag{136}$$

582 where the trace is taken over the $h = -1/5$ representation. We can split the trace up into
583 the sum of traces over level subspaces of the representation, i.e. spaces where the descendent
584 states have the same $L_0$ eigenvalue. Let $\mathcal{H}_N$ denote the subspace at level $N$. (We are only
585 working in the $h = -1/5$ representation so won't add an additional label to $\mathcal{H}$ to represent
586 this.) The trace (136) is given by

$$\text{Tr}\Big(e^{-R(I_1(L)-\alpha I_5(L)+\beta_9\alpha^2 J_9(L)+\alpha_{13}\alpha^3 I_{13}(L)+\beta_{13}\alpha^3 J_{13}(L)+\dots)}\Big) = \sum_{N=0}^{\infty} \text{Tr}_{\mathcal{H}_N}\Big(e^{-\frac{R}{L}Q\left(\frac{\alpha}{L^4}\right)}\Big)\,, \tag{137}$$

587 where $\frac{1}{L}Q\left(\frac{\alpha}{L^4}\right)$ is defined by its asymptotic expansion

$$\frac{1}{L}Q\left(\frac{\alpha}{L^4}\right) \sim I_1(L) - \alpha I_5(L) + \beta_9\alpha^2 J_9(L) + \alpha_{13}\alpha^3 I_{13}(L) + \beta_{13}\alpha^3 J_{13}(L) + \dots\,. \tag{138}$$

588 If the space $\mathcal{H}_N$ has dimension $n$ then we will label the $n$ eigenvalues of $Q$ by $q_i$, $i = 1,\dots,n$
589 and we find

$$\text{Tr}_{\mathcal{H}_N}\Big(e^{-\frac{R}{L}Q\left(\frac{\alpha}{L^4}\right)}\Big) = \sum_{i=1}^{n} e^{-\frac{R}{L}q_i\left(\frac{\alpha}{L^4}\right)} \tag{139}$$

590 It is the eigenvalues $\frac{1}{L}q_i\left(\frac{\alpha}{L^4}\right)$ that will be found by solving the TBA equations (97) and plugging
591 the solutions into (98). In our numerical analysis we have only solved the one particle excited
592 state TBA equation (115) and hence have only found one of the eigenvalues $q_i$ at each level.
593 The others can be obtained by solving the TBA equations (97) with more than one pole.

594      We will verify the above claims that the TBA is encoding the spectrum of the transformed
595 GGE with some numerical tests in the next section.

## 5.5   Numerical results

597 In the previous section we found asymptotic solutions to the TBA equations (95), (96) for the
598 ground state and (97), (98) for the excited states. The energy (96) and (98) are then given
599 as asymptotic expansions in $\alpha$

$$E(L) \sim \mathcal{E}_0(L) + \alpha\mathcal{E}_1(L) + \alpha^2\mathcal{E}_2(L) + \alpha^3\mathcal{E}_3(L) + O(\alpha^4)\,, \tag{140}$$

600 where the $\mathcal{E}_k$ can be read off from (109) for the ground state and are given in (131) for
601 the excited states. As explained in section 5.4 the coefficients $\epsilon_k$ are expected to be linear
602 combinations of the eigenvalues of the charges $I_{2n-1}(L)$ and $J_{2n-1}(L)$ for levels 0, 1, 2 and 3
603 where we have only one state. The exact relations are

$$\mathcal{E}_0(L) = \mathcal{I}_1(L)\,, \tag{141}$$

$$\mathcal{E}_1(L) = -\mathcal{I}_5(L)\,, \tag{142}$$

$$\mathcal{E}_2(L) = \beta_9\mathcal{J}_9(L)\,, \tag{143}$$

$$\mathcal{E}_3(L) = \alpha_{13}\mathcal{I}_{13}(L) + \beta_{13}\mathcal{J}_{13}(L)\,, \tag{144}$$

where the numerical constants $\beta_9, \alpha_{13}$ and $\beta_{13}$ are given in (59–61) and as before $\mathcal{I}_{2n-1}(L)$ and $\mathcal{J}_{2n-1}(L)$ are eigenvalues of $I_{2n-1}(L)$ and $J_{2n-1}(L)$.

However for levels 4 and 5 in the $h = -1/5$ representation we have two states. Hence we need to find the eigenvalues of the operator $Q$ defined in (138). We can find the elements of the matrix $Q$ up to $O(\alpha^4)$ using (138) and the explicit expressions for the charges $I_{2n-1}$ and $J_{2n-1}$ given in appendix B.3. This then allows us to compute the eigenvalues up to $O(\alpha^4)$. For level 4 the two eigenvalues of $\frac{1}{L} Q \left( \frac{\alpha}{L^4} \right)$ are

$$\frac{1}{L} q_1 \left( \frac{\alpha}{L^4} \right) = \frac{239}{60} \frac{1}{L} + \left( \frac{29871991}{756000} + \frac{2\sqrt{5149}}{5} \right) \frac{\alpha}{L^4} + \left( \frac{65155161071}{21600000} + \frac{1581671\sqrt{5149}}{40650} \right) \left( \frac{\alpha}{L^4} \right)^2$$
$$+ \left( \frac{906057445994257}{2592000000} + \frac{400124699794729\sqrt{5149}}{83722740000} \right) \left( \frac{\alpha}{L^4} \right)^3 + O(\alpha^4), \quad (145)$$

$$\frac{1}{L} q_2 \left( \frac{\alpha}{L^4} \right) = \frac{239}{60} \frac{1}{L} + \left( \frac{29871991}{756000} - \frac{2\sqrt{5149}}{5} \right) \frac{\alpha}{L^4} + \left( \frac{65155161071}{21600000} - \frac{1581671\sqrt{5149}}{40650} \right) \left( \frac{\alpha}{L^4} \right)^2$$
$$+ \left( \frac{906057445994257}{2592000000} - \frac{400124699794729\sqrt{5149}}{83722740000} \right) \left( \frac{\alpha}{L^4} \right)^3 + O(\alpha^4), \quad (146)$$

and for level 5 the two eigenvalues are

$$\frac{1}{L} q_1 \left( \frac{\alpha}{L^4} \right) = \frac{299}{60} \frac{1}{L} + \left( \frac{99483211}{756000} + \frac{2\sqrt{36409}}{5} \right) \frac{\alpha}{L^4} + \left( \frac{511399295771}{21600000} + \frac{558565553\sqrt{36409}}{5461350} \right) \left( \frac{\alpha}{L^4} \right)^2$$
$$+ \left( \frac{16846422773011117}{2592000000} + \frac{2527183186828313923\sqrt{36409}}{79536916860000} \right) \left( \frac{\alpha}{L^4} \right)^3 + O(\alpha^4), \quad (147)$$

$$\frac{1}{L} q_2 \left( \frac{\alpha}{L^4} \right) = \frac{299}{60} \frac{1}{L} + \left( \frac{99483211}{756000} - \frac{2\sqrt{36409}}{5} \right) \frac{\alpha}{L^4} + \left( \frac{511399295771}{21600000} - \frac{558565553\sqrt{36409}}{5461350} \right) \left( \frac{\alpha}{L^4} \right)^2$$
$$+ \left( \frac{16846422773011117}{2592000000} - \frac{2527183186828313923\sqrt{36409}}{79536916860000} \right) \left( \frac{\alpha}{L^4} \right)^3 + O(\alpha^4). \quad (148)$$

In tables 1 – 4 we collect our numerical results and compare them to the expected analytic values up to level 5[2]. In all cases we have good numerical agreement which supports out claim that the TBA equations (97) and (98) give the spectrum of the transformed GGE.

We note that for levels 4 and 5 where we have two eigenvalues the TBA equations give the eigenvalue corresponding to the positive square root. We believe that the other root can be obtained by solving the TBA equation (97) with two poles but we have not verified this.

---

[2]In all of our numerical results we have not done a serious error analysis, even though errors do arise from discretising and introducing cut-offs to our integration range in the TBA. This is because our results were in such agreement with the known analytic values that we did not feel the need to perform such an analysis.

$\mathcal{E}_0(2\pi)$ numerical and analytic values

| Level | Numerical value | Analytic value |
|---|---|---|
| 0 | $-0.01666666666666666$ | $-\frac{1}{60} = -0.016666666666666666$ |
| 1 | $0.9833333333333341$ | $\frac{59}{60} = 0.9833333333333333$ |
| 2 | $1.983333333333334$ | $\frac{119}{60} = 1.9833333333333333$ |
| 3 | $2.983333333333334$ | $\frac{179}{60} = 2.9833333333333333$ |
| 4 | $3.983333333333334$ | $\frac{239}{60} = 3.9833333333333333$ |
| 5 | $4.983333333333334$ | $\frac{299}{60} = 4.9833333333333333$ |

Table 1: We list the numerical values of $\mathcal{E}_0(2\pi)$ ($L = 2\pi$) when the TBA equations are solved for levels 0 to 5. In the final column we list the analytic results that come from diagonalising the charges directly.

$\mathcal{E}_1(2\pi)$ numerical and analytic values

| Level | Numerical value | Analytic value |
|---|---|---|
| 0 | $0.00011772486772486771$ | $\frac{89}{756000} = 0.00011772486772486772$ |
| 1 | $0.07821560846561151$ | $\frac{59131}{756000} = 0.07821560846560846$ |
| 2 | $2.1565489417989436$ | $\frac{1630351}{756000} = 2.156548941798942$ |
| 3 | $16.234882275132286$ | $\frac{12273571}{756000} = 16.234882275132275$ |
| 4 | $68.2158287299221$ | $\frac{29871991}{756000} + \frac{2\sqrt{5149}}{5} = 68.21582872992198$ |
| 5 | $207.91611903557663$ | $\frac{99483211}{756000} + \frac{2\sqrt{36409}}{5} = 207.91611903557674$ |

Table 2: We list the numerical values of $\mathcal{E}_1(2\pi)$ ($L = 2\pi$) when the TBA equations are solved for levels 0 to 5. In the final column we list the analytic results that come from diagonalising the charges directly.

$\mathcal{E}_2(2\pi)$ numerical and analytic values

| Level | Numerical value | Analytic value |
|---|---|---|
| 0 | $-0.00008004629629629623$ | $-\frac{1729}{21600000} = -0.0000800462962962963$ |
| 1 | $0.023933842592585384$ | $\frac{516971}{21600000} = 0.023933842592592593$ |
| 2 | $11.574614398148247$ | $\frac{250011671}{21600000} = 11.574614398148148$ |
| 3 | $438.0852949537006$ | $\frac{9462642371}{21600000} = 438.0852949537037$ |
| 4 | $5808.453146384214$ | $\frac{65155161071}{21600000} + \frac{1581671\sqrt{5149}}{40650} = 5808.45314638423$ |
| 5 | $43191.34083200923$ | $\frac{511399295771}{21600000} + \frac{558565553\sqrt{36409}}{5461350} = 43191.34083200952$ |

Table 3: We list the numerical values of $\mathcal{E}_2(2\pi)$ ($L = 2\pi$) when the TBA equations are solved for levels 0 to 5. In the final column we list the analytic results that come from diagonalising the charges directly.

$\mathcal{E}_3(2\pi)$ numerical and analytic values

| Level | Numerical value | Analytic value |
|---|---|---|
| 0 | $-0.00041588850308641904$ | $-\frac{1077983}{2592000000} = -0.00041588850308641975$ |
| 1 | $0.01004061612654321$ | $\frac{26025277}{2592000000} = 0.01004061612654321$ |
| 2 | $86.7328804540904$ | $\frac{224811626137}{2592000000} = 86.73288045408951$ |
| 3 | $16554.39302029211$ | $\frac{42908986708597}{2592000000} = 16554.393020292053$ |
| 4 | $692495.4337312711$ | $\frac{906057445994257}{2592000000} + \frac{400124699794729\sqrt{5149}}{83722740000} = 692495.4337312633$ |
| 5 | $12562179.006709663$ | $\frac{168464227730111117}{2592000000} + \frac{2527183186828313923\sqrt{36409}}{79536916860000} = 12562179.006709557$ |

Table 4: We list the numerical values of $\mathcal{E}_3(2\pi)$ $(L = 2\pi)$ when the TBA equations are solved for levels 0 to 5. In the final column we list the analytic results that come from diagonalising the charges directly.

## 5.6   Non-asymptotic solutions to the TBA

In section 5.4 we found the solutions to the TBA equations that reproduced the asymptotic results from section 4. Here we will show that there are additional solutions to the TBA equations that one needs to consider when calculating the transformed GGE. We will again restrict to the $h = -\frac{1}{5}$ sector in the transformed GGE. These additional solutions therefore correspond to states in the $H_{D,-\frac{1}{5}}$ defect Hilbert space. (the defect Hilbert spaces were introduced in (12).) We will begin by recalling the one particle excited state TBA equations

$$\epsilon(\theta) = e^\theta + \frac{\alpha C}{L^4}e^{5\theta} + \log\left(\frac{S(\theta - \eta)}{S(\theta - \bar{\eta})}\right) - \int_{-\infty}^{\infty}\varphi(\theta - \theta')\log\left(1 + e^{-\epsilon(\theta')}\right)\frac{d\theta'}{2\pi}\,, \qquad (149)$$

$$E(L) = \frac{i}{L}\left(e^{\bar{\eta}} - e^\eta\right) - \frac{1}{L}\int_{-\infty}^{\infty}e^\theta\log\left(1 + e^{-\epsilon(\theta)}\right)\frac{d\theta}{2\pi}\,, \qquad (150)$$

and the constraint

$$2n\pi i = e^\eta + \frac{\alpha C}{L^4}e^{5\eta} - \log S(2i\mathrm{Im}(\eta)) - \int_{-\infty}^{\infty}\varphi(\eta - \theta')\log\left(1 + e^{-\epsilon(\theta')}\right)\frac{d\theta'}{2\pi}\,. \qquad (151)$$

In order to solve (149) and (151) to find solutions that were missed in the asymptotic analysis we will choose alternative expansions to (101) and (118) for $\epsilon$ and $\eta$. We assume now $\epsilon(\theta)$ has the expansion

$$\epsilon(\theta) = \sum_{n=0}^{\infty}\epsilon_{\frac{n}{4}}(\theta)\left(\frac{\alpha}{L^4}\right)^{\frac{n}{4}}\,. \qquad (152)$$

and $\eta$ can be expanded as

$$\eta = -\frac{1}{4}\log\left(\frac{\alpha}{L^4}\right) + \sum_{n=0}^{\infty}\eta_{\frac{n}{4}}\left(\frac{\alpha}{L^4}\right)^{\frac{n}{4}}\,. \qquad (153)$$

The leading order $-\frac{1}{4}\log\left(\frac{\alpha}{L^4}\right)$ term for $\eta$ can be determined as follows. Assume that as $\alpha \to 0$ the pseudo energy tends to a finite, $\alpha$ independent function of $\theta$, $\epsilon(\theta) \to \epsilon_0(\theta)$. We will further assume that in this limit $e^\eta \to \left(\frac{\alpha}{L^4}\right)^\nu e^{\eta_0}$ for some $\nu$ and $\eta_0$ to be determined. We plug both of these limits into the constraint equation (151) to determine the power $\nu$.

635       We first note that if $\mathrm{Re}(\nu) \neq 0$ then in the limit $\alpha \to 0$ the kernel $\varphi(\eta - \theta')$ vanishes so we
636       drop the convolution term. If $\mathrm{Re}(\nu) = 0$ then $\varphi(\eta - \theta')$ oscillates without decaying as $\alpha \to 0$,
637       so we don't have a well defined limit. We now need to determine the behaviour of the driving
638       term $\log S(2i\mathrm{Im}(\eta))$ as $\alpha \to 0$. If $\mathrm{Im}(\nu) = 0$ then we have $\log S(2i\mathrm{Im}(\eta)) \to \log S(2i\mathrm{Im}(\eta_0))$.
639       However if $\mathrm{Im}(\nu) \neq 0$ then $\log S(2i\mathrm{Im}(\eta))$ oscillates without decaying as $\alpha \to 0$ so we again
640       don't have a well defined limit. Hence we must have $\nu \in \mathbb{R} \setminus \{0\}$.

641       Using both of these limits in the constraint equation (151) gives the leading order terms

$$2n\pi i \approx \left(\frac{\alpha}{L^4}\right)^\nu e^{\eta_0} + \left(\frac{\alpha}{L^4}\right)^{5\nu+1} C e^{5\eta_0} - \log S(2i\mathrm{Im}(\eta_0)) \,. \tag{154}$$

642       If $\nu > 0$ then both $\alpha^\nu$ and $\alpha^{5\nu+1}$ are subleading and we have

$$2n\pi i \approx -\log S(2i\mathrm{Im}(\eta_0)) \,. \tag{155}$$

643       However this equation has no solutions for finite $\eta_0$, hence $\nu < 0$. Now the $\alpha^\nu$ term diverges
644       as $\alpha \to 0$ and so the $\alpha^{5\nu+1}$ term must also diverge at the same rate in order for them to cancel.
645       This fixes $\nu = -\frac{1}{4}$ and hence we have the leading $\eta$ behaviour from (153)

$$e^\eta \sim \left(\frac{\alpha}{L^4}\right)^{-\frac{1}{4}} e^{\eta_0} \Rightarrow \eta \sim -\frac{1}{4}\log\left(\frac{\alpha}{L^4}\right) + \eta_0 \,. \tag{156}$$

646       As in section 5.4 we will expand the TBA equations as an asymptotic series in $\alpha$ and solve
647       them term by term. First we need to expand the terms in the TBA equations. We start with
648       $\log\left(\frac{S(\theta-\eta)}{S(\theta-\bar\eta)}\right)$

$$\log\left(\frac{S(\theta-\eta)}{S(\theta-\bar\eta)}\right) = 4\sqrt{3}\,\mathrm{Im}\left(e^{-\eta_0}\right)e^\theta \left(\frac{\alpha}{L^4}\right)^{\frac{1}{4}} - 4\sqrt{3}\,\mathrm{Im}\left(\eta_{\frac{1}{4}}e^{-\eta_0}\right)e^\theta \left(\frac{\alpha}{L^4}\right)^{\frac{1}{2}} + O\left(\alpha^{\frac{3}{4}}\right) \,. \tag{157}$$

649       Next we provide the expansion of $S(2i\mathrm{Im}(\eta))$

$$\log S(2i\mathrm{Im}(\eta)) = \log S(2i\mathrm{Im}(\eta_0)) - 2\left(\frac{\alpha}{L^4}\right)^{\frac{1}{4}} \mathrm{Im}\left(\eta_{\frac{1}{4}}\right)\varphi(2i\mathrm{Im}(\eta_0)) \tag{158}$$

$$- 2\left(\frac{\alpha}{L^4}\right)^{\frac{1}{2}}\left(\mathrm{Im}\left(\eta_{\frac{1}{2}}\right)\varphi(2i\mathrm{Im}(\eta_0)) + i\,\mathrm{Im}\left(\eta_{\frac{1}{4}}\right)^2 \varphi'(2i\mathrm{Im}(\eta_0))\right) + O(\alpha^{\frac{3}{4}}) \,,$$

650       and finally $\varphi(\eta - \theta')$

$$\varphi(\eta - \theta') = -2\sqrt{3}e^{-\eta_0}e^{\theta'}\left(\frac{\alpha}{L^4}\right)^{\frac{1}{4}} + 2\sqrt{3}\eta_{\frac{1}{4}}e^{-\eta_0}e^{\theta'}\left(\frac{\alpha}{L^4}\right)^{\frac{1}{2}} + O(\alpha^{\frac{3}{4}}) \,. \tag{159}$$

651       If we plug (157) into the non-linear integral equation (149) then we get the series of equations

$$\epsilon_0(\theta) = e^\theta - \int_{-\infty}^\infty \varphi(\theta - \theta') L(\epsilon_0(\theta'))\frac{d\theta'}{2\pi} \,, \tag{160}$$

$$\epsilon_{\frac{1}{4}}(\theta) = 4\sqrt{3}\,\mathrm{Im}\left(e^{-\eta_0}\right)e^\theta - \int_{-\infty}^\infty \varphi(\theta - \theta')\epsilon_{\frac{1}{4}}(\theta')L'(\epsilon_0(\theta'))\frac{d\theta'}{2\pi} \,, \tag{161}$$

$$\epsilon_{\frac{1}{2}}(\theta) = -4\sqrt{3}\,\mathrm{Im}\left(\eta_{\frac{1}{4}}e^{-\eta_0}\right)e^\theta - \int_{-\infty}^\infty \varphi(\theta-\theta')\left(\epsilon_{\frac{1}{2}}(\theta')L'(\epsilon_0(\theta')) + \frac{1}{2}\epsilon_{\frac{1}{4}}(\theta')^2 L''(\epsilon_0(\theta'))\right)\frac{d\theta'}{2\pi} \,, \tag{162}$$

and if we plug (158) and (159) into the constraint (151) we have the series of equations

$$0 = e^{\eta_0} + C e^{5\eta_0} \,, \tag{163}$$

$$2n\pi i = \left( e^{\eta_0} + 5C e^{5\eta_0} \right) \eta_{\frac{1}{4}} - \log S(2i\mathrm{Im}(\eta_0)) \,, \tag{164}$$

$$0 = \frac{1}{2} \left( e^{\eta_0} + 25C e^{5\eta_0} \right) \eta_{\frac{1}{4}}^2 + \left( e^{\eta_0} + 5C e^{5\eta_0} \right) \eta_{\frac{1}{2}} + 2\mathrm{Im}\left( \eta_{\frac{1}{4}} \right) \varphi(2i\mathrm{Im}(\eta_0))$$

$$+ \int_{-\infty}^{\infty} 2\sqrt{3} e^{-\eta_0} e^{\theta} L(\epsilon_0(\theta)) \frac{d\theta}{2\pi} \,. \tag{165}$$

We note that (160) doesn't contain $\eta_{\frac{n}{4}}$ and hence can be solved by itself to find $\epsilon_0$. Similarly (163) can be solved to find

$$\eta_0 = \frac{1}{4} \log(-1/C) + \frac{\pi i k}{2} \,, \quad k = 0, 1, 2, 3 \,, \tag{166}$$

where we recall that $C$ defined in (87) is negative so we can choose the branch cut such that $\log(-1/C) \in \mathbb{R}$. (We are ignoring the solution $e^{\eta_0} = 0$.) We can then solve (164) to find four possible values for $\eta_1$ and use all the previous solutions to solve (165) for $\eta_{\frac{1}{2}}$. While so far we have been able to solve each of the equations independently we note that equations coming from higher orders in $\alpha$ will again have to be solved in tandem as we did for the asymptotic solutions in section 5.4.

We can continue to solve the series of equations coming from the integral equation (160) and the constraint (151) iteratively to find an asymptotic solution to $\eta$ and $\epsilon$. There will be four possible solutions, which when added to the asymptotic solution gives us five in total for each $n \in \mathbb{Z}$ in the constraint.

However we do not want to include all of these solutions in the transformed GGE (14). We only want solutions $\epsilon(\theta)$ and $\eta$ such that when they are plugged into the integral (150) for $E(L)$ we have

$$\mathrm{Re}(E(L) - E_0(L)) > 0 \,, \tag{167}$$

where $E_0(L)$ is the ground state energy. This is to ensure the convergence of the GGE (14) which is a sum over the exponentials $e^{-R(E(L)-E_0(L))}$.

Based on the results for free fermion GGEs [15–17] we conjecture that if we add these terms to the GGE then we will have the full modular transformation. This conjecture can also be extended to the case with a finite number of charges inserted as was done for free fermions in [16].

A non-trivial check of the conjecture would be to verify that with these additional terms inserted the expression for the transformed GGE is real. We believe that the individual energies $E(L)$ that come from solving (97) and plugging the solution into (98) will have branch points in $\alpha$ on the negative real line. This is for both the asymptotic solutions from section 5.4 and the ones from this section. Hence the energies may individually be complex, but by including all of them in the transformed expression for the GGE we get a real quantity.

In order to verify this we would like to numerically determine the branch points of the energies. This could be done by solving the TBA equations (97) numerically for fixed values of $\alpha$ and finding where the energies (98) become complex. This would give exact solutions for a given $\alpha$ to the TBA equations rather than the power series solutions we have discussed so far. However, so far we have not been able to find a stable numerical algorithm to solve (97) for $\alpha \neq 0$. We leave it to future work to find the solutions to (95) for fixed values of $\alpha$ and determine there branch points.

We will end this section with a brief discussion on the large $\alpha$ behaviour of the solutions to the TBA equations (149) and the constraint (151). Since the solutions only depend on the

combination $\frac{\alpha}{L^4}$, the large $\alpha$ limit is equivalent to the small $L$ limit. We will assume that the pseudo energy $\epsilon(\theta)$ and the pole $\eta$ have the leading behaviour

$$\epsilon(\theta) \sim \left(\frac{\alpha}{L^4}\right)^\mu \epsilon_0(\theta), \quad e^\eta \sim \left(\frac{\alpha}{L^4}\right)^\nu e^{\eta_0} \tag{168}$$

As was discussed above for the $\alpha \to 0$ limit, the constraint equation (151) again only has a well defined limit if $\nu \in \mathbb{R} \setminus \{0\}$. Note that the value of $\mu$ does not change the fact that the convolution term is suppressed in (151) as $\alpha \to -\infty$. Hence we have

$$2n\pi i \approx \left(\frac{\alpha}{L^4}\right)^\nu e^{\eta_0} + \left(\frac{\alpha}{L^4}\right)^{5\nu+1} C e^{5\eta_0} - \log S(2i\mathrm{Im}(\eta_0)). \tag{169}$$

In the limit $\alpha \to -\infty$ this equation only has solutions if $\nu = -\frac{1}{5}$. Then the $\alpha^\nu$ term is subleading and the leading order constraint equation is

$$2n\pi i \approx C e^{5\eta_0} - \log S(2i\mathrm{Im}(\eta_0)). \tag{170}$$

For $\nu \in \mathbb{R} \setminus \{0\}$ the $\log\left(\frac{S(\theta-\eta)}{S(\theta-\bar{\eta})}\right)$ term in (149) tends to 0 as $\alpha \to -\infty$. The integral in (149) is also subleading and hence we have the leading order behaviour

$$\epsilon(\theta) = \frac{\alpha}{L^4} C e^{5\theta}. \tag{171}$$

So as $\alpha \to \infty$ we have

$$\epsilon(\theta) \sim \frac{\alpha}{L^4} C e^{5\theta}, \quad e^\eta \sim \left(\frac{\alpha}{L^4}\right)^{-\frac{1}{5}} e^{\eta_0}. \tag{172}$$

If we use these limits in the energy integral (150) then we find the leading order behaviour of the spectrum is

$$E(L) \sim (\alpha L)^{-\frac{1}{5}} \left( i\left(e^{\bar{\eta}_0} - e^{\eta_0}\right) + \int_{-\infty}^{\infty} e^\theta \log\left(1 + e^{C e^{5\theta}}\right) \frac{d\theta}{2\pi} \right). \tag{173}$$

# 6 Conclusions and Outlook

Let us begin our conclusion with a brief summary of the results presented in this paper. We will just focus on the main example from the paper, the Lee-Yang model where the $I_5(R)$ KdV charge was inserted into the characters, with chemical potential $\alpha$, to give us our GGE

$$\mathrm{Tr}_{\mathcal{H}_i}\left(e^{-L(I_1(R)+\alpha I_5(R))}\right). \tag{174}$$

We expanded the GGE as an asymptotic series in $\alpha$

$$\mathrm{Tr}_{\mathcal{H}_i}\left(e^{-L(I_1(R)+\alpha I_5(R))}\right) = \sum_{n=1}^{\infty} \frac{(-\alpha L)^n}{n!} \mathrm{Tr}_{\mathcal{H}_i}\left(I_5(R)^n e^{-L I_1(R)}\right), \tag{175}$$

and took the modular transform of each term. The expressions for the transformed correlators can be written as correlators of the original KdV charges as well as the correlators of the zero modes of the other quasi-primary fields present in the theory. For example

$$\mathrm{Tr}\left(I_5(R)^2 e^{-L I_1(R)}\right) = \left(\frac{R}{L}\right)^2 \mathrm{Tr}\left(I_5(L)^2 e^{-R I_1(L)}\right) - \frac{412776}{116875} \frac{R}{L^2} \mathrm{Tr}\left(J_9(L) e^{-R I_1(L)}\right), \tag{176}$$

709 where $J_9(L)$ is the zero mode on the cylinder of the quasi-primary field at level 10 in the $h = 0$
710 representation. Once we have transformed each term we can then resum them into a GGE
711 with all charges from the quasi-primary fields present, not just the subset of the KdV charges

$$\text{Tr}\left(e^{-L(\alpha I_5(R)+I_1(R))}\right) \sim \text{Tr}\left(e^{-R(I_1(L)+\alpha_5 I_5(L)+\beta_9 J_9(L)+\alpha_{13} I_{13}(L)+\beta_{13} J_{13}(L)+\dots)}\right) , \quad (177)$$

712 where the $\alpha_{2n-1}$ and $\beta_{2n-1}$ are given in (59–61). Based on the results for the free fermion
713 model [15–17] we assume that the expressions (177) only match asymptotically and that as
714 a GGE the right hand side is a formal expression that diverges.

715 In order to find a regularised expression for the right hand side of (177) we turned to the
716 TBA. If the transformed GGE is just given as a trace over the $h = -1/5$ representation then the
717 TBA equations that give the ground state energy are

$$\epsilon(\theta) = e^\theta + \frac{\alpha C}{L^4} e^{5\theta} - \int_{-\infty}^{\infty} \varphi(\theta - \theta') \log\left(1 + e^{-\epsilon(\theta')}\right) \frac{d\theta'}{2\pi} , \quad (178)$$

$$E_0(L) = -\frac{1}{L} \int_{-\infty}^{\infty} e^\theta \log\left(1 + e^{-\epsilon(\theta)}\right) \frac{d\theta}{2\pi} , \quad (179)$$

718 and the TBA equations for the excited states are

$$\epsilon(\theta) = e^\theta + \frac{\alpha C}{L^4} e^{5\theta} + \sum_{i=1}^{N} \log\left(\frac{S(\theta - \theta_i)}{S(\theta - \bar{\theta}_i)}\right) - \int_{-\infty}^{\infty} \varphi(\theta - \theta') \log\left(1 + e^{-\epsilon(\theta')}\right) \frac{d\theta'}{2\pi} , \quad (180)$$

$$E(L) = \frac{i}{L} \sum_{i=1}^{N} \left(e^{\bar{\theta}_i} - e^{\theta_i}\right) - \frac{1}{L} \int_{-\infty}^{\infty} e^\theta \log\left(1 + e^{-\epsilon(\theta)}\right) \frac{d\theta}{2\pi} . \quad (181)$$

719 and the poles $\theta_i$ satisfy the constraints

$$2 n_i \pi i = e^{\theta_i} + \frac{\alpha C}{L^4} e^{5\theta_i} - \log S(\theta_i - \bar{\theta}_i) + \sum_{\substack{j=1 \\ j \neq i}}^{N} \log\left(\frac{S(\theta_i - \theta_j)}{S(\theta_i - \bar{\theta}_j)}\right) - \int_{-\infty}^{\infty} \varphi(\theta_i - \theta') \log\left(1 + e^{-\epsilon(\theta')}\right) \frac{d\theta'}{2\pi} , \quad (182)$$

720 where $n_i \in \mathbb{Z}$.

721 If we assume that both the pseudo energy $\epsilon(\theta)$ and the poles $\theta_i$ have asymptotic expansions
722 as a power series in $\alpha$ then we can reproduce the spectrum of the GGE on the right hand side
723 of (177). We verified this for the case with one pole but conjecture that all the other states
724 can also be obtained this way.

725 We then found another set of solutions to the TBA equations which had the leading be-
726 haviour $\alpha^{-1/4}$ as $\alpha \to 0^+$. When exponentiated these energies have a vanishing asymptotic
727 expansion and where hence missed in the original asymptotic analysis. However we conjecture
728 that they should be included in the full expression for the transformed GGE and they are the
729 only additional terms that we have to add to the asymptotic results. Hence the full spectrum
730 of the transformed GGE is contained in the above TBA equations.

731 It is also worth noting that these TBA equations can be written as the same Y system that
732 one has for the ordinary Lee-Yang model. The original derivation of Y systems from TBA
733 equations was given in [28], and in [29] Castro-Alvaredo showed that the same Y system also
734 encodes the TBA equations for GGEs. For the case of our Lee-Yang TBA equations (178) and
735 (180), we define

$$Y(\theta) = e^{\epsilon(\theta)} . \quad (183)$$

736 Then $Y(\theta)$ satisfies the Y system

$$Y\left(\theta - \frac{i\pi}{3}\right) Y\left(\theta + \frac{i\pi}{3}\right) = 1 + Y(\theta) . \quad (184)$$

As was noted in [28] the functions $Y(\theta)$ are periodic $Y(\theta) = Y\left(\theta + \frac{5\pi i}{3}\right)$. Hence we can further define

$$t(\lambda) = Y\left(\frac{5}{3}\log\lambda\right), \tag{185}$$

which satisfies the T system

$$t\left(e^{\frac{i\pi}{5}}\lambda\right) t\left(e^{-\frac{i\pi}{5}}\lambda\right) = 1 + t(\lambda). \tag{186}$$

This is the same T system first derived in [3]. However our function $t(\lambda)$ also has a dependence on $\alpha$ and hence has different analytic properties to the one defined in [3]. In [3] the asymptotic expansion of $t(\lambda)$ as $\lambda \to \infty$ gave the eigenvalues of the KdV charges in the theory. It would be interesting to understand if the asymptotic expansion of our function $t(\lambda)$ as $\lambda \to \infty$ again contains the eigenvalues of higher spin conserved charges that are present in the theory represented by our transformed GGE.

While we have provided evidence for our conjecture that the spectrum for the transformed GGE is fully encoded in the TBA equations (178) and (180) we have not provided a rigorous proof of this statement. In [17] it was proven that for free fermions the full spectrum of the transformed GGE is encoded in the TBA equations for that model. However the proof required having the explicit expressions for the GGEs and then using Poisson summation to perform the modular transformation. Here we do not have an explicit expression for the original GGE and hence cannot attempt to use the same methods.

In section 3 we saw that when we found an asymptotic expression for the transformed GGE we had not only KdV charges appearing in the expression, but the zero modes of the other quasi-primary fields were also present. These are also conserved charges and so physically they should also be inserted into the GGE if we want to consider the most general GGEs used to describe a physical system. It would be interesting to study these GGEs and their modular properties. We can repeat the analysis of section 3 to find an asymptotic expression for the transformed GGE in terms of a new GGE. However we do not have TBA equations that encode the spectrum of these charges that are not KdV charges, hence we can't reproduce the analysis of section 5 even though we would again expect there to be terms missing from the asymptotic results. We leave the study of these more general GGEs to future work.

Naturally we would like to extend these results to other models where there are interesting GGEs to study. We can naturally extend the results of this paper to the case of minimal models where again the KdV charges are inserted into the characters to give us our GGEs.

An interesting point to mention is that in a generic 2d CFT, there exist further infinite sets of commuting conserved charges that are independent to the KdV hierarchy. In particular there exist hierarchies that are related to the ZMS-Bullough-Dodd model, see for example [30], and can be constructed by considering certain integrable perturbations of CFTs [31] (in fact there are two sets of Bullough-Dodd charges which depend on the choice of the integrable perturbation). In the case of the Lee-Yang Model that we have analysed in this paper, the KdV hierarchy and the Bullough-Dodd hierarchies exactly coincide. It is then natural to ask about GGEs with Bullough-Dodd charges inserted in them in a more general setting.

There is also the $BO_2$ hierarchy that exists for CFTs that have a $U(1)$ current. GGEs with these charges inserted have been studied in [32, 33]. Studying their modular properties is an open question that would be interesting to explore.

Finally we mention GGEs arising from W algebras. The $W_3$ algebra contains a weight 3 primary field with zero mode $W_0$. This zero mode commutes with the stress tensor zero mode $L_0$, hence we can construct a GGE

$$\text{Tr}(e^{\alpha W_0} q^{L_0 - \frac{c}{24}}). \tag{187}$$

The modular properties of this GGE is still an open question. The first few terms in the asymptotic expansion and their modular transforms were calculated in [23, 34]. The additional

charges and their thermal correlators have recently been calculated in [35, 36]. Putting these results together could allow us to find an asymptotic expression for the modular transform of (187) similar to our results in section 3. If TBA equations for the additional charges are known then we may hope to repeat the arguments of section 5 to find the full modular transform of (187).

# Acknowledgements

We would like to thank O. Castro-Alvaredo and A. Konechny for useful discussions during the completion of this work. We would especially like to thank G. M. T. Watts for many extensive and valuable discussions that aided in the progression of this work, reading early drafts of the paper and suggesting useful edits.

FK would like to acknowledge the funding from the STFC grant ST/Y509279/1 in supporting this work. MD would like to thank EPSRC for support under grant EP/V520019/1 while the majority of this work was carried out and support from the French Agence Nationale de la Recherche (ANR) under grant ANR-21-CE40-0003 (project CONFICA) while this work was written up.

Throughout this work we have made extensive use of the Virasoro Mathematica notebook by Matthew Headrick, which can be found at https://sites.google.com/view/matthew-headrick/mathematica.

# A  Modular forms

In this appendix we will list the relevant facts about modular forms that appear in this paper. Proofs of the following statements can be found in [37] and most of the notation will be the same.

The modular group will be denoted by

$$\Gamma_1 = \mathrm{SL}(2, \mathbb{Z})/\{\pm I\}\,, \tag{188}$$

Consider a matrix

$$\begin{pmatrix} a & b \\ c & d \end{pmatrix} \in \Gamma_1\,. \tag{189}$$

If a holomorphic function $f(\tau)$, defined in the upper half plane, has the following transformation property

$$f\left(\frac{a\tau + b}{c\tau + d}\right) = (c\tau + d)^k f(\tau)\,, \tag{190}$$

then we say that the function is a holomorphic modular form of weight $k$ on $\Gamma_1$. We will denote the space of modular forms of weight $k$ on $\Gamma_1$ by $M_k(\Gamma_1)$.

The group $\Gamma_1$ is finitely generated by the matrices

$$\pm \begin{pmatrix} 1 & 1 \\ 0 & 1 \end{pmatrix}, \quad \pm \begin{pmatrix} 0 & 1 \\ -1 & 0 \end{pmatrix}, \tag{191}$$

hence we only need to check that a function transforms as a modular form under

$$T : \tau \mapsto \tau + 1\,, \quad S : \tau \mapsto \frac{-1}{\tau}\,, \tag{192}$$

to verify it is an element of $M_k(\Gamma_1)$.

An important fact about the space $M_k(\Gamma_1)$ is that it is finite dimensional. The space $M_{2k}(\Gamma_1)$ is generated by the Eisenstein series, which we now define.

The Eisenstein series $E_{2k}(\tau)$ are elements of $M_{2k}(\Gamma_1)$ for $k = 2, 3, \ldots$ and they are defined by

$$E_{2k}(\tau) = 1 + \frac{2}{\zeta(1-2k)} \sum_{n=0}^{\infty} \frac{n^{2k-1}q^n}{1-q^n} \,, \quad q = e^{2\pi i\tau} \,. \tag{193}$$

For $k = 1$ the Eisenstein series $E_2(\tau)$ is quasi-modular which means that under a modular transform we have the transformation property

$$E_2\left(\frac{a\tau+b}{c\tau+d}\right) = (c\tau+d)^2 E_2(\tau) - \frac{6i}{\pi}c(c\tau+d) \,. \tag{194}$$

We also encounter quasi-modular forms. For our purpose we will define the space of quasi-modular forms of weight $k$ and depth $p$, denoted by $\widetilde{M}_k^{(\leq p)}(\Gamma_1)$, to be

$$\widetilde{M}_k^{(\leq p)}(\Gamma_1) = \bigoplus_{r=0}^{p} M_{k-2r}(\Gamma_1) \cdot E_2^r \,, \tag{195}$$

where the coefficient of $E_2^p$ is non-zero.

Finally we define the Serre derivative. The Serre derivative acting on a modular form $f(\tau)$ of weight $k$ is defined to be

$$D_k f(\tau) = \frac{1}{2\pi i} \frac{d}{d\tau} f(\tau) - \frac{k}{12} E_2(\tau) f(\tau) \,. \tag{196}$$

By using the transformation of $\frac{d}{d\tau}$ under a modular transform we can see that $D_k f(\tau)$ is a modular form of weight $k + 2$.

# B  Construction of Charges

In this appendix we will explain how to construct the charges used throughout this paper. These charges are the zero modes of quasi-primary fields on the cylinder so we will begin by explaining how we use the algorithm of Gaberdiel in [38] to map fields from the cylinder to the plane. We will then apply this map to the case of the quasi-primary field at level 6 that is linearly independent from the quasi-primary field that gives the KdV charge $I_5$. Finally we will discuss the charges in the Lee-Yang minimal model and give explicit expressions for all the charges used in this work.

## B.1  Mapping between the cylinder and plane

To start we will explain how to map a field from the cylinder to the plane. Suppose that we have a field $\phi^{\mathrm{pl}}(z)$ defined on the complex plane $z \in \mathbb{C}$ We will assume that this field is a level $N$ descendent of a primary fields of weight $h$, hence the field $\phi^{\mathrm{pl}}$ has weight $h + N$. We want an expression for this field on the cylinder of circumference $R$ with coordinate $w \sim w + iR$. We will denote the field on the cylinder by $\phi^{\mathrm{cyl}}(w)$ and we will find an expression for it in terms of fields on the plane, i.e. an expression of the form

$$\phi^{\mathrm{cyl}}(w) = \Phi(z) \,, \tag{197}$$

where $\Phi(z)$ is constructed out of fields defined on the plane. The conformal map that we use to map between the cylinder and the plane is

$$z = e^{\frac{2\pi}{R}w} \,. \tag{198}$$

In order to find the expression $\phi^{\mathrm{cyl}}(w) = \Phi(z)$ we first find the asymptotic state associated to $\phi^{\mathrm{pl}}(z)$

$$|\phi\rangle = \lim_{z \to 0} \phi^{\mathrm{pl}}(z)|0\rangle . \tag{199}$$

We then find an intermediate state $|\Phi\rangle$ by acting on $|\phi\rangle$ with Virasoro modes $L_n$

$$|\Phi\rangle = z^{L_0} \prod_{n=1}^{\infty} e^{R_n L_n} |\phi\rangle, \tag{200}$$

where the product is written in ascending order of $n$

$$\prod_{n=1}^{\infty} e^{R_n L_n} = e^{R_1 L_1} \times e^{R_2 L_2} \times e^{R_3 L_3} \times \dots . \tag{201}$$

The algorithm for computing the $R_n$ is given in [38], we have listed the relevant ones for our calculations in table 5. We note that for $n$ odd and greater than 1 $R_n = 0$ so we have not listed them in table 5. (For a general conformal map $z = f(w)$ the $R_n$ will be functions of $w$ and $z^{L_0}$ becomes $f'(z)$.) Although we have an infinite product and the exponentials also contain infinite products these expressions can truncate to a finite one since any operator of the form $L_{n_1} \dots L_{n_i}$ with $n_1 + \dots + n_i > N$ will annihilate $|\phi\rangle$.

The intermediate state $|\Phi\rangle$ will be of the form

$$|\Phi\rangle = \sum_{m=0}^{N} z^{m+h} |\Phi_m\rangle , \tag{202}$$

so we can then use the state operator correspondence to find the fields $\Phi_m(z)$ corresponding to $|\Phi_m\rangle$. We then define the field

$$\Phi(z) = \sum_{m=0}^{N} z^{m+h} \Phi_m(z) . \tag{203}$$

This field gives the an expression for the field $\phi^{\mathrm{cyl}}(w)$ on the cylinder in terms of fields on the plane

$$\phi^{\mathrm{cyl}}(w) = \Phi(z) . \tag{204}$$

Our charges are the zero modes of fields on the cylinder. If we have a field $\phi^{\mathrm{cyl}}(w)$ on the cylinder of circumference $R$, we integrate it on a spatial slice to obtain the associated charge $\phi_0(R)$. We can then use our map (204) to express this as an integral on the plane

$$\phi_0(R) = \int_0^{iR} \frac{dw}{2\pi i} \phi^{\mathrm{cyl}}(w) = \frac{R}{2\pi} \oint \frac{dz}{2\pi i\, z} \Phi(z). \tag{205}$$

## B.2 Example of a Weight 6 Field

As an explicit example we will apply the algorithm of the previous section to the weight 6 quasi-primary field

$$\phi^{\mathrm{pl}}(z) = (T'T')(z) - \frac{4}{5}(T''T)(z) - \frac{1}{42} T^{(4)}(z), \tag{206}$$

which is defined on the plane. This field is linearly independent to the quasi-primary field that gives the KdV charge $I_5$.

| $n$ | $R_n(w)$ |
|---|---|
| 0 | $w$ |
| 1 | $\frac{1}{2}\left(\frac{2\pi}{R}\right)^1$ |
| 2 | $-\frac{1}{12}\left(\frac{2\pi}{R}\right)^2$ |
| 4 | $-\frac{1}{48}\left(\frac{2\pi}{R}\right)^4$ |
| 6 | $\frac{1}{12096}\left(\frac{2\pi}{R}\right)^6$ |
| 8 | $-\frac{1}{138240}\left(\frac{2\pi}{R}\right)^8$ |
| 10 | $\frac{1}{2280960}\left(\frac{2\pi}{R}\right)^{10}$ |
| 12 | $-\frac{389}{13586227200}\left(\frac{2\pi}{R}\right)^{12}$ |
| 14 | $\frac{1}{464486400}\left(\frac{2\pi}{R}\right)^{14}$ |

Table 5: Table of some of the $R_n$'s necessary for the map of a field from the cylinder to plane

First we need its associated asymptotic state which is

$$|\phi\rangle = \left(L_{-3}^2 - \frac{8}{5}L_{-4}L_{-2} - \frac{4}{7}L_{-6}\right)|0\rangle. \tag{207}$$

Next, acting on the state to generate the intermediate state $|\Phi\rangle$ as in (200) yields

$$\left(\frac{2\pi}{R}\right)^6\left(z^6\left(L_{-3}^2 - \frac{8}{5}L_{-4}L_{-2} - \frac{4}{7}L_{-6}\right)|0\rangle + z^4\left(\frac{4}{5}L_{-2}^2 + \frac{14c-95}{210}L_{-4}\right)|0\rangle\right.$$
$$\left. + z^3\frac{70c+29}{420}L_{-3}|0\rangle + z^2\frac{280c-163}{2100}L_{-2}|0\rangle + \frac{31c}{16800}|0\rangle\right). \tag{208}$$

Finding the state defined in (203) gives

$$\Phi(z) = \left(\frac{2\pi}{R}\right)^6\left(z^6\left((T'T')(z) - \frac{4}{5}(T''T)(z) - \frac{1}{42}T^{(4)}(z)\right) + z^4\left(\frac{4}{5}(TT)(z) + \frac{14c-95}{420}T''(z)\right)\right.$$
$$\left. + z^3\frac{70c+29}{420}T'(z) + z^2\frac{280c-163}{2100}T(z) + \frac{31c}{16800}\right). \tag{209}$$

The brackets denote normal ordering as defined in [39]. This then gives use the field $\phi^{\text{cyl}}(w)$ on the cylinder

$$\phi^{\text{cyl}}(w) = \left(\frac{2\pi}{R}\right)^6\left(z^6\left((T'T')(z) - \frac{4}{5}(T''T)(z) - \frac{1}{42}T^{(4)}(z)\right) + z^4\left(\frac{4}{5}(TT)(z) + \frac{14c-95}{420}T''(z)\right)\right.$$
$$\left. + z^3\frac{70c+29}{420}T'(z) + z^2\frac{280c-163}{2100}T(z) + \frac{31c}{16800}\right). \tag{210}$$

We can then integrate this as in (205) to obtain the conserved quantity

$$J_5(R) = \phi_0(R) = \left(\frac{2\pi}{R}\right)^5\left(-\frac{18}{5}\sum_{k=1}^{\infty}k^2 L_{-k}L_k - \frac{3}{100}L_0 + \frac{31c}{16800}\right). \tag{211}$$

### B.3 Charges in the Lee-Yang Model

In this section we will present the charges relevant to the Lee-Yang model. These will include the KdV charges which have been calculated previously (see for example [3]). However we will find new simpler expressions for them by using a bases of states in the Lee-Yang theory which already has null states removed. We will also calculate the charges associated with the other quasi-primary fields in the theory. While these don't commute with the KdV charges they still appear when we take the modular transform of a GGE as see in section 4.

There is natural basis of states in the vacuum module which avoids the null vectors in the Lee-Yang model [40], that is the vacuum module is given by

$$\mathcal{H}_0 = \text{span}\left\{L_{-n_1}...L_{-n_m}|0\rangle \mid m \geq 0, n_m > 1, n_i > n_{i+1} + 1\right\}. \tag{212}$$

We can then use this bases of states when calculating the quasi-primary fields. For example at level 4 we have a single state

$$L_{-4}|0\rangle. \tag{213}$$

If we act on this state with $L_1$ we obtain

$$L_1 L_{-4}|0\rangle = 5L_{-3}|0\rangle \neq 0. \tag{214}$$

Hence we have no quasi-primary states at level 4. This means we have no $I_3$ KdV charge as has previously been pointed out in [14].

We can also find the quasi-primary state at level 6. A generic state at level 6 is

$$(aL_{-6} + bL_{-4}L_{-2})|0\rangle, \tag{215}$$

for constants $a$ and $b$. When we act with $L_1$ we obtain

$$(7aL_{-5} + 5bL_{-3}L_{-2})|0\rangle \tag{216}$$

However the term $L_{-3}L_{-2}$ is not in $\mathcal{H}_0$. We can exchange it for terms in $\mathcal{H}_0$ by using the null states in the Lee-Yang model. In the vacuum sector there is a null state at level 4 given by

$$\left(L_{-4} - \frac{5}{3}L_{-2}^2\right)|0\rangle = 0. \tag{217}$$

Hence we can act on this with $L_{-1}$ to obtain the relation

$$L_{-3}L_{-2}|0\rangle = \frac{2}{5}L_{-5}|0\rangle. \tag{218}$$

Using this in (216) we obtain

$$L_1(aL_{-6} + bL_{-4}L_{-2})|0\rangle = (7a + 2b)L_{-5}|0\rangle \in \mathcal{H}_0. \tag{219}$$

Hence we have the quasi-primary state at level 6

$$\left(L_{-6} - \frac{7}{2}L_{-4}L_{-2}\right)|0\rangle. \tag{220}$$

This is proportional to the state which gives the KdV charge $I_5$. However if we map this state to the cylinder then we get an expression for the zero mode which only contains quadratic and linear terms in the Virasoro modes rather than the usual expression for $I_5$ which contains terms that are cubic (see for example the expression in [3]). Hence using the representation (212) for the vacuum module leads to simpler expressions for the charges.

898 Using the representation (212) for the vacuum model we have calculated the zero modes
899 of all the quasi-primary fields with even weight up to weight 14

$$I_1(R) = \left(\frac{2\pi}{R}\right)^1 \left(L_0 + \frac{11}{60}\right),$$

$$I_3(R) = 0,$$

$$I_5(R) = \left(\frac{2\pi}{R}\right)^5 \left(\frac{1}{5}\sum_{k=1}^{\infty}(k^2+6)L_{-k}L_k + \frac{3}{5}L_0^2 + \frac{73}{600}L_0 + \frac{341}{756000}\right),$$

$$I_7(R) = \left(\frac{2\pi}{R}\right)^7 \left(\frac{1}{28}\sum_{k=1}^{\infty}(13k^4+82k^2-546)L_{-k}L_k - \frac{39}{4}L_0^2 + \frac{90137}{35280}L_0 - \frac{5863}{8467200}\right),$$

$$J_9(R) = \left(\frac{2\pi}{R}\right)^9 \left(-\frac{55}{27216}\sum_{k=1}^{\infty}(17k^6+30054)L_{-k}L_k - \frac{275495}{9072}L_0^2 - \frac{7934443}{1306368}L_0 - \frac{5797}{78382080}\right),$$

$$J_{13}(R) = \left(\frac{2\pi}{R}\right)^{13} \left(-\frac{23}{8895744}\sum_{k=1}^{\infty}(19k^{10}+51294138)L_{-k}L_k - \frac{196627529}{2965248}L_0^2 - \frac{3864911011991}{291424573440}L_0 - \frac{1494977}{1589588582400}\right),$$

$$I_{13}(R) = \left(\frac{2\pi}{R}\right)^{13} \left(-\frac{91}{211612500000}\sum_{k=1}^{\infty}(1631557057290 - 18646489477k^2 - 14982597630k^4 - 275953986k^6\right.$$

$$\left. - 4754750k^8 + 546098k^{10})L_{-k}L_k - \frac{637}{937500}\mathcal{L}(5,3,0) - \frac{45227}{3375000}\mathcal{L}(4,2,0) - \frac{637}{8437500}\mathcal{L}(6,2,0) - \frac{4949056407113L_0^2}{14107500000}\right.$$

$$\left. - \frac{187569810221381L_0}{3047220000000} - \frac{825517}{174960000000}\right).$$

(221)

900 where the $\mathcal{L}(l,n,m)$ are defined by

$$\mathcal{L}(l,n,m) = (T^{(l)}(T^{(n)}T^{(m)}))_0 = \oint \frac{dz}{2\pi i}z^{l+n+m+5}(T^{(l)}(T^{(n)}T^{(m)}))(z),$$

$$= \sum_{\substack{i\leq-2\\j\leq-2}}(-1)^{l+n+m}\prod_{a=2}^{l+1}(i+a)\prod_{b=2}^{n+1}(j+b)\prod_{c=2}^{m+1}(c-i-j)L_iL_jL_{-i-j}$$

$$+ \sum_{\substack{i\leq-2\\j\geq-1}}(-1)^{l+n+m}\prod_{a=2}^{l+1}(i+a)\prod_{b=2}^{n+1}(j+b)\prod_{c=2}^{m+1}(c-i-j)L_iL_{-i-j}L_j$$

$$+ \sum_{\substack{i\geq-1\\j\leq-2}}(-1)^{l+n+m}\prod_{a=2}^{l+1}(i+a)\prod_{b=2}^{n+1}(j+b)\prod_{c=2}^{m+1}(c-i-j)L_jL_{-i-j}L_i$$

$$+ \sum_{\substack{i\geq-1\\j\geq-1}}(-1)^{l+n+m}\prod_{a=2}^{l+1}(i+a)\prod_{b=2}^{n+1}(j+b)\prod_{c=2}^{m+1}(c-i-j)L_{-i-j}L_jL_i .$$

(222)

901 The $I_{2n-1}$ are the KdV charges coming from a weight $2n$ quasi-primary field. They can be
902 uniquely fixed (up to a factor) by imposing that they all commute [3]. $J_9$ is the zero mode of
903 the unique quasi-primary field at level 10, it does not commute with the KdV charges. $J_{13}$ is
904 the zero mode of the quasi-primary field at level 14 that is linearly independent to the field
905 that gives the KdV charge $I_{13}$, it also doesn't commute with the KdV charges.

## C  Eigenvalues and Thermal Correlation Functions of Charges

In this appendix we explain how to compute the thermal correlation functions of the charges using the techniques of [14]. These thermal correlation functions are given by modular differential operators acting on the characters of the theory.

### C.1  Eigenvalues

First we will give an expression for the correlators in terms of $k^{\text{th}}$ power sums (225) of the eigenvalues. We will explain how to calculate these power sums using the characteristic equation of a matrix which avoids us having to explicitly find the eigenvalues we are summing over.

Consider an operator $\mathcal{O}$ with scaling dimension $h_i$. We want to calculate the thermal correlator

$$\langle \mathcal{O}^k \rangle_i(\tau) \,, \tag{223}$$

where $\langle \ldots \rangle_i$ is defined in (26).

In order to do this we need to find the sums of powers of the eigenvalues at each level. Consider restricting the operator to the level $N$ subspace and let us denote the dimension of this subspace $n$. We can obtain the eigenvalues of $\mathcal{O}$ in this subspace and label them $\lambda_i$ for $i = 1, \ldots, n$. Then the thermal correlator is

$$\langle \mathcal{O}^k \rangle_i(\tau) = q^{h_i - \frac{c}{24}} \sum_{N=0}^{\infty} p_k(\lambda_1, \ldots, \lambda_n) q^N \,, \quad q = e^{2\pi i \tau} \,, \tag{224}$$

where $p_k(\lambda_1, \ldots, \lambda_n)$ is the $k^{\text{th}}$ power sum

$$p_k(\lambda_1, \ldots, \lambda_n) = \sum_{i=1}^{n} \lambda_i^k \,. \tag{225}$$

In order to calculate the correlator we need to know $p_k(\lambda_1, \ldots, \lambda_n)$. We could find the eigenvalues at each level and then sum their powers. However for level subspaces with $n > 4$ we won't, in general, be able to find eigenvalues since they will be roots of polynomials of order greater than 4.

Instead we can calculate $p_k(\lambda_1, \ldots, \lambda_n)$ using Newton's identities. These identities relate the coefficients in the characteristic equation of $\mathcal{O}$ restricted to a level subspace, to the power sum of the eigenvalues. It is much easier to compute the characteristic polynomial for a matrix than finding it's eigenvalues, especially when $n > 4$.

We start by defining the coefficients $x_i$ in the characteristic polynomial

$$\det(\lambda I - \mathcal{O}) = \prod_{i=1}^{n}(\lambda - \lambda_i) = \sum_{i=0}^{n} x_{n-i}(\lambda_1, \ldots, \lambda_n) \lambda^i. \tag{226}$$

We then define the elementary symmetric polynomials $e(\lambda_1, \ldots, \lambda_n)$

$$\begin{aligned}
e_0(\lambda_1, \ldots, \lambda_n) &= 1 \\
e_1(\lambda_1, \ldots, \lambda_n) &= \lambda_1 + \cdots + \lambda_n \\
e_2(\lambda_1, \ldots, \lambda_n) &= \sum_{1 \le i < j \le n} \lambda_i \lambda_j \\
&\vdots \\
e_n(\lambda_1, \ldots, \lambda_n) &= \lambda_1 \ldots \lambda_n \\
e_k(\lambda_1, \ldots, \lambda_n) &= 0 \,, \quad k > n.
\end{aligned} \tag{227}$$

933   We have the relation

$$x_i(\lambda_1, \ldots, \lambda_n) = (-1)^i e_i(\lambda_1, \ldots, \lambda_n). \tag{228}$$

934   Newton's identity then states the following relation between the $x_i$ and the $p_k$

$$p_k(\lambda_1, \ldots, \lambda_n) = -k x_k(\lambda_1, \ldots, \lambda_n) - \sum_{i=1}^{k-1} x_{k-i}(\lambda_1, \ldots, \lambda_n) p_i(\lambda_1, \ldots, \lambda_n), \quad n \geq k \geq 1,$$

$$\tag{229}$$

$$p_k(\lambda_1, \ldots, \lambda_n) = -\sum_{i=k-n}^{k-1} x_{k-i}(\lambda_1, \ldots, \lambda_n) p_i(\lambda_1, \ldots, \lambda_n), \quad k > n \geq 1.$$

935   Hence we can use the coefficients of the characteristic polynomial to compute the thermal
936   correlator of $\mathcal{O}^k$.

937       As an example we have tabulated in Tables 6 and 7 the necessary sums of eigenvalues of
938   the composite operator $I_5 J_{13}$ which we have calculated using this method. This can then be
939   used to calculate the thermal correlator (248). In general however, the tables become rather
940   large and are not very illuminating to have in the document.

| Level $m$ | $\Lambda_m = \sum \lambda_i$ |
|:---:|:---:|
| 0 | $-\frac{509787157}{12017289682294400000}$ |
| 2 | $-\frac{2569135573534504727}{13219018651238400000}$ |
| 3 | $-\frac{50731836772785663329 27}{13219018651238400000}$ |
| 4 | $-\frac{981246408964814147312327}{13219018651238400000}$ |
| 5 | $-\frac{56558651194472972584841327}{13219018651238400000}$ |
| 6 | $-\frac{10971963472538794035413 4161}{9442156179456 00000}$ |
| 7 | $-\frac{1247009833113290085767 3903327}{6609509325619200000}$ |

Table 6: Sums of powers of eigenvalues $I_5 J_{13}$ in $h = 0$ representation.

| Level $m$ | $\Lambda_m = \sum \lambda_i$ |
|:---:|:---:|
| 0 | $-\frac{1141081727}{13219018651238400000}$ |
| 1 | $-\frac{18487172153927}{13219018651238400000}$ |
| 2 | $-\frac{614331959543590361}{1888431235891200000}$ |
| 3 | $-\frac{6214449825743713137527}{13219018651238400000}$ |
| 4 | $-\frac{548022831223321427788127}{6609509325619200000}$ |
| 5 | $-\frac{235173539304715601912537 9}{508423794278400000}$ |

Table 7: Sums of powers of eigenvalues $I_5 J_{13}$ in $h = -\frac{1}{5}$ representation.

## C.2   Thermal Correlation Functions

942   Using the method presented in the previous section to calculate sums of eigenvalues of oper-
943   ators, we can find the expression of thermal correlation functions in terms of modular differ-
944   ential operators acting on the characters of the CFT as was done in [14].

945       We will do this explicitly for the charge $J_5 = J_5(2\pi)$ which we defined in section B.2. We
946   can calculate it's thermal correlator in a generic rational CFT as follows. We first note that in

947  the vacuum sector $h_0 = 0$ we have

$$\langle J_5 \rangle_0 = \frac{31c}{16800} + 0q + O(q^2) \,, \tag{230}$$

948  and in the other representations $h_i$ with $i \neq 0$ we have

$$\langle J_5 \rangle_{i \neq 0} = \frac{31c - 504h}{16800} + \frac{31c - 121464h - 504}{16800} q + O(q^2) \,, \tag{231}$$

949  From [14] we know that the thermal correlator must be a weight 6 modular differential oper-
950  ator acting on the characters. We only have a linear $L_0$ term in (211), hence we must have a
951  first order differential operator. Using the definition of the Serre derivative $D$ and Eisenstein
952  series $E_{2k}$ in appendix A the only first order weight 6 differential operator we can construct is

$$\langle J_5 \rangle_i = \left( a\, E_4 D + b\, E_6 \right) \chi_i \,, \tag{232}$$

953  where $a, b$ are constants. Then by performing a $q$-series expansion of the above differential
954  operator and comparing with the leading order terms in (230) and (231), we deduce

$$\langle J_5 \rangle_i = \left( -\frac{3}{100} E_4 D + \frac{c}{1680} E_6 \right) \chi_i \,. \tag{233}$$

955  If we restrict to the Lee-Yang model we can find simpler expressions for the correlation func-
956  tions compared to those given in [14] for generic CFTs. This is because the characters satisfy
957  a second order differential equation which can be used to reduce the order of differential
958  operator acting on the characters.

959      We will demonstrate this for the case of the correlator $\langle I_5 \rangle$ in the Lee-Yang model. (For
960  the rest of this section we will drop the $i$ subscript since the specific representation won't be
961  important.) From [14], $\langle I_5 \rangle$ in the Lee-Yang model is

$$\langle I_5 \rangle = \left( D^3 - \frac{1}{720} E_4 D + \frac{11}{9450} E_6 \right) \chi \,. \tag{234}$$

962  However we also have the modular differential equation satisfied by the two characters of
963  Lee-Yang [21]

$$\left( D^2 - \frac{11}{3600} E_4 \right) \chi = 0 \,. \tag{235}$$

964  Acting on this with the Serre derivative $D_4$ and using $D_4 E_4 = -\frac{1}{3} E_6$ we find

$$\left( D^3 - \frac{11}{3600} E_4 D + \frac{11}{10800} E_6 \right) \chi = 0 \,. \tag{236}$$

965  Using this to eliminate the $D^3$ term in (234) gives use the first order differential operator

$$\left( \frac{1}{600} E_4 D + \frac{11}{75600} E_6 \right) \chi = 0 \,. \tag{237}$$

966  Using the differential equation (235) means that all thermal correlators will be first order
967  differential operators acting on the characters.

968      For example if we want the thermal correlator $\langle I_5^2 \rangle$ in the Lee-Yang model we know it must
969  be a weight 12, depth 1, first order modular differential operator acting on the characters so
970  we have the ansatz

$$\langle I_5^2 \rangle = \left( a_1 E_4 E_6 D + a_2 E_4^3 + a_3 E_6^2 \right) \chi + E_2 \left( a_4 D + a_5 E_4 E_6 \right) \chi \,, \tag{238}$$

971  which has five constants $a_i$ that can be fixed by finding the $2^{\text{nd}}$ power sum of the eigenvalues
972  as detailed in section C.1.

973      In the following subsections we will give all the thermal correlation functions, in the Lee-
974  Yang model, that are relevant to this work. They were all computed using the techniques
975  detailed in section C.1 this section.

### C.2.1 One Point Functions

$$\langle I_5 \rangle = \left( \tfrac{1}{600} E_4 D + \tfrac{11}{75600} E_6 \right) \chi, \tag{239}$$

$$\langle J_9 \rangle = \left( -\tfrac{187}{1306368} E_4^2 D - \tfrac{187}{3919104} E_4 E_6 \right) \chi, \tag{240}$$

$$\langle I_{13} \rangle = \left( -\tfrac{19747}{5832000000} E_4^2 E_6 - \tfrac{5341}{1188000000} E_4^3 D - \tfrac{889}{320760000} E_6^2 D \right) \chi, \tag{241}$$

$$\langle J_{13} \rangle = \left( -\tfrac{437}{582266880} E_4^2 E_6 - \tfrac{3059}{4625786880} E_4^3 D - \tfrac{10925}{29142457344} E_6^2 D \right) \chi. \tag{242}$$

### C.2.2 Two Point Functions

$$\langle I_5^2 \rangle = \left( \tfrac{977}{22680000} E_4 E_6 D + \tfrac{3937}{432000000} E_4^3 + \tfrac{5669}{1143072000} E_6^2 \right) \chi$$
$$+ E_2 \left( -\tfrac{91}{2160000} E_4^2 D - \tfrac{91}{6480000} E_4 E_6 \right) \chi, \tag{243}$$

$$\langle I_5 J_9 \rangle = \left( -\tfrac{6827183}{296284262400} E_6^2 E_4 - \tfrac{16388119}{940584960000} E_4^4 - \tfrac{93101129}{1283898470400} E_6 E_4^2 D \right) \chi$$
$$+ E_2 \left( \tfrac{76109}{1881169920} E_6 E_4^2 + \tfrac{28985}{1069915392} E_6^2 D + \tfrac{8789}{194088960} E_4^3 D \right) \chi \tag{244}$$

### C.2.3 Three Point Functions

$$\langle I_5^3 \rangle = \left( \tfrac{236364271}{86416243200000} E_6^3 + \tfrac{494225369}{32659200000000} E_4^3 E_6 + \tfrac{1157429}{86400000000} E_4^4 D + \tfrac{21351661}{1143072000000} E_4 E_6^2 D \right) \chi$$
$$+ E_2 \left( -\tfrac{7974967}{518400000000} E_4^4 - \tfrac{474617}{23328000000} E_4 E_6^2 - \tfrac{497867}{7776000000} E_4^2 E_6 D \right) \chi$$
$$+ E_2^2 \left( \tfrac{37037}{2073600000} E_4^2 E_6 + \tfrac{2303}{115200000} E_4^3 D + \tfrac{31}{2592000} E_6^2 D \right) \chi \tag{245}$$

### C.2.4 Four Point Functions

$$\langle I_5 J_9 I_5 J_9 + 2 I_5^2 J_9^2 \rangle = \big( \tfrac{217939227481921982970761 3}{3922170296936693760000000 0} E_4^8 + \tfrac{3261128141852799871003297969}{7516682487267296123289600000} E_4^5 E_6^2$$
$$+ \tfrac{4443955492489611413324 9}{2891031725872036970496 00} E_4^2 E_6^4 + \tfrac{1797258671670302467637935 7}{8030643682977880473600000 0} E_4^6 E_6 D$$
$$+ \tfrac{13447381664688693623851}{48183862097867282841600} E_4^3 E_6^3 D + \tfrac{365114027354495}{34835065137266688} E_6^5 D \big) \chi$$
$$+ E_2 \big( -\tfrac{1845971371461482486256988 7}{2130578936300254003200000 00} E_4^6 E_6 - \tfrac{2145587212938780676961401}{20879673575742489231360 00} E_4^3 E_6^3$$
$$- \tfrac{7991249766071003993}{22586185358375288832 0} E_6^5 - \tfrac{3676618312680240690922 3}{1821007637863464960000 00} E_4^7 D$$
$$- \tfrac{18949790853757962690931 9}{167037388605939913850880} E_4^4 E_6^2 D - \tfrac{13629984197781552899}{66922030691482337280} E_4 E_6^4 \big) \chi$$
$$+ E_2^2 \big( \tfrac{113655918391381520043 01}{42718374662661734400000} E_4^7 + \tfrac{11329892784121632558550 9}{79541613621876149452800} E_4^4 E_6^2 +$$
$$\tfrac{3741640654381407688889}{15659755181806866923520} E_4 E_6^4 + \tfrac{86626897675944766897}{96624895070306304000} E_4^5 E_6 D$$
$$+ \tfrac{4198392142384509586 33}{65248979924195278848 0} E_4^2 E_6^3 D \big) \chi$$
$$+ E_2^3 \big( -\tfrac{420831892527650263}{1095342940068249600} E_4^5 E_6 - \tfrac{17371382461645795}{670897550791802 88} E_4^2 E_6^3$$
$$- \tfrac{655898828464786511}{6102624951808819200} E_4^6 D - \tfrac{71634855640572155}{1868928891491450 88} E_4^3 E_6^2 D$$
$$- \tfrac{25790460875702875}{1144718946038513664} E_6^4 D \big) \chi \tag{246}$$

$$\langle I_5^3 I_{13}\rangle = (-\frac{3662789242251036625469}{5744286720000000000000000}E_4^8 - \frac{829919046794601603749}{2983837824000000000000000}E_4^6 E_6 D - \frac{43752429748829672779547}{878875868160000000000000}E_4^5 E_6^2$$
$$- \frac{234794331598895297207 9}{676734418483200000000000}E_4^3 E_6^3 D - \frac{10253251682424762132791}{5813763867878400000000000}E_4^2 E_6^4 - \frac{1755294025516129867}{1346345316771840000000000}E_6^5 D)\chi$$
$$+ E_2(\frac{449152119628129528793}{179030269440000000000000}E_4^7 D + \frac{11360417540096992 77559}{80563621248000000000000}E_6^2 E_4^4 D + \frac{9522101518506256241}{375963565824000000000000}E_6^4 E_4 D$$
$$+ \frac{3640771544831927121763}{366198278400000000000000}E_6 E_4^6 + \frac{1554345511673424702041}{131831380224000000000000}E_6^3 E_4^3 + \frac{15731532887841983 8243}{3875842578585600000000000}E_6^5)\chi$$
$$+ E_2^2(-\frac{99734751071982908147}{895151347200000000000 0}E_6 E_4^5 D - \frac{38670103939681218767}{4833817274880000000000}E_6^3 E_4^2 D - \frac{52306420490432254657}{1713208320000000000000}E_4^7$$
$$- \frac{1436431475895774703849}{878875868160000000000000}E_6^2 E_4^4 - \frac{7227523086579850439}{263662760448000000000000}E_6^4 E_4)\chi$$
$$+ E_2^3(\frac{37749886664319446 89}{282679372800000000000 0}E_4^6 D + \frac{2992912386847019}{628176384000000000000}E_6^2 E_4^3 D + \frac{233020899293}{831409920000000000}E_6^4 D$$
$$+ \frac{377646503248171703}{85660416000000000000}E_6 E_4^5 + \frac{1017904924658167}{342641664000000000}E_6^3 E_4^2)\chi$$
$$\tag{247}$$

$$\langle I_5^3 J_{13}\rangle = (-\frac{17873436145260800359}{33955446260736000000000}E_6 E_4^6 D - \frac{49039152240800944980248 9}{7475196419051500339200000}E_6^3 E_4^3 D - \frac{92716955395499288947}{3767498995201956170956800}E_6^5 D$$
$$- \frac{612351928167536196113}{43164142534656000000000}E_4^8 - \frac{7169575623957618748413 73}{647203153164632064000000}E_6^2 E_4^5 - \frac{3361152009340424011942883}{85624977163680822067200000}E_6^4 E_4^2)\chi$$
$$+ E_2(\frac{4915080143718468157}{10348326479462400000000}E_4^7 D + \frac{2260654561876999456183}{84752793866797056000000 0}E_6^2 E_4^4 D + \frac{79483870998598392829}{166115475979822229760000}E_6^4 E_4 D$$
$$+ \frac{1014541232963208145 31}{4586190144307200000000}E_6 E_4^6 + \frac{1273701667950551073824 3}{485402364873474048000000}E_6^3 E_4^3 + \frac{39680387596385790613}{4391024466993234984960000}E_6^5)\chi$$
$$+ E_2^2(-\frac{1329564636918812861}{6306011448422240000000}E_6 E_4^5 D - \frac{1398099098325038689}{9245759330923315200000}E_6^3 E_4^2 D - \frac{45313164080668629943}{6670822028083200000000}E_4^7$$
$$- \frac{480391290954347056469}{13208227615604736000000 0}E_6^2 E_4^4 - \frac{96714650888817229}{1584987313872568320 00}E_6^4 E_4)\chi$$
$$+ E_2^3(\frac{3091899079658220 79}{12229840384819200000000}E_4^6 D + \frac{145460634198308051}{16143389307961344000 0}E_6^2 E_4^3 D + \frac{159846756287}{3021489977425920}E_6^4 D$$
$$+ \frac{12584290522849297}{12828503900160000000}E_6 E_4^5 + \frac{1492883098488757}{2257816686428160000}E_6^3 E_4^2)\chi$$
$$\tag{248}$$

### C.2.5 Six Point Functions

$$\langle I_5^6\rangle = (\frac{4933344922206498844523471}{5643509760000000000000000}E_4^9 + \frac{7787739236115473312156044 1}{82959593472000000000000000}E_4^6 E_6^2$$
$$+ \frac{16247612446364320656611277 1}{2822285369917440000000000}E_4^3 E_6^4 + \frac{51469469692192836453088181}{3733883544400773120000000 00}E_6^6$$
$$+ \frac{135302164964875254677089}{32920473600000000000000}E_4^7 E_6 D + \frac{10088010610841244 65663}{1269789696000000000000}E_4^4 E_6^3 D$$
$$+ \frac{43878417952324626225 8293}{49389993973555200000000000}E_4 E_6^5 D)\chi$$
$$+ E_2(-\frac{13460764140934194041924 29}{50791587840000000000000}E_4^7 E_6 - \frac{54998516624011085098765 37}{11199545118720000000000000}E_4^4 E_6^3$$
$$- \frac{1042524100253555760517}{2015918121369600000000}E_4 E_6^5 - \frac{47635867931274538969 3}{89579520000000000000}E_4^8 D$$
$$- \frac{256102757295680787853 01}{5925685248000000000000}E_4^5 E_6^2 D - \frac{10869082560559813828551 7}{67197270712320000000000}E_4^2 E_6^4 D)\chi$$
$$+ E_2^2(\frac{2778703440861548647 37}{199065600000000000000}E_4^8 + \frac{664282069414669217454 37}{60949905408000000000000}E_4^5 E_6^2$$
$$+ \frac{691953340111094055949}{17919272189952000000 00}E_4^2 E_6^4 + \frac{68230028806013854126 1}{1209323352000000000000}E_4^6 E_6 D$$
$$+ \frac{1250699504032655425 9}{17777055744000000000}E_4^3 E_6^3 D + \frac{59177887440790376 7}{2239909023744000000 0}E_6^5 D)\chi$$
$$+ E_2^3(-\frac{2632129661468288897993}{3627970560000000000000}E_4^6 E_6 - \frac{13111503925652433941}{15237476352000000000}E_4^3 E_6^3$$
$$- \frac{3318117971641012397}{11199545118720000000 0}E_6^5 - \frac{68392401230782519891}{4031078400000000000 00}E_4^7 D$$
$$- \frac{13834166805095898541}{14511882240000000000}E_4^4 E_6^2 D - \frac{25352801672559097}{148142131200000000 0}E_4 E_6^4 D)\chi$$
$$+ E_2^4(\frac{4310775931268934211 7}{3869835264000000000000}E_4^7 + \frac{3461798088323165845 1}{58047528960000000000}E_4^4 E_6^2$$
$$+ \frac{120977896233403}{1209323352000000 0}E_4 E_6^4 + \frac{22495159155372463}{597196800000000000}E_4^5 E_6 D + \frac{653851361661301}{2418647040000000000}E_4^2 E_6^3 D)\chi$$
$$+ E_2^5(-\frac{19212872091349937}{199065600000000000000}E_4^5 E_6 - \frac{58267207067069}{895795200000000000}E_4^2 E_6^3$$
$$- \frac{43687413654877461 7}{161243136000000000000}E_4^6 D - \frac{194747805703493}{2015553920000000 0}E_4^3 E_6^2 D - \frac{114470161877}{201553920000000 00}E_6^4 D)\chi$$
$$\tag{249}$$

### C.3  Modular transform of correlators

What's actually important here is the modular transformation of the $\langle I_5^n \rangle$ thermal correlators. When performing the modular transformation of these quasi-modular forms, we pick up additional pieces which we can then rewrite in terms of other thermal correlators. For example, the following transformation was derived by finding $\langle I_5^3 \rangle$ as a modular differential operator acting on the characters of the theory, (245), then taking the $S : \tau \mapsto -\frac{1}{\tau}$ transformation and noticing that the result can be written in terms of the other thermal correlation functions (241), (242), (244)

$$\langle I_5^3 \rangle\left(-\tfrac{1}{\tau}\right) = \tau^{18}\langle I_5^3 \rangle(\tau) - \tfrac{6i}{\pi}\tau^{17}\left(\tfrac{103194}{116875}\langle I_5 J_9 \rangle(\tau)\right) - \tfrac{36}{\pi^2}\tau^{16}\left(-\tfrac{45}{16}\langle I_{13} \rangle(\tau) - \tfrac{31941}{2875}\langle J_{13} \rangle(\tau)\right). \quad (250)$$

This is crucial for the re-exponentiation of the GGE after we take the modular $S$ transformation of it.

# D  Numerical algorithm for TBA

In this appendix we will briefly outline our approach to solving the TBA equations numerically. We do this by discretising the integrals into finite sums and then setting up iteration schemes. Our iteration scheme for finding the pseudo energy $\epsilon(\theta)$ is the same as the one used in equations (2.2) of [25] with $a = 1$. We will also explicitly give the iteration scheme we used to solve for the poles $\eta$ in the excited states.

### D.1  Ground State

Let us start with the ground state TBA equations (95) and (96). In order to solve the TBA equations we expanded $\epsilon(\theta)$ as an asymptotic series in $\alpha$ (101) and then solved the TBA equation (95) order by order in $\alpha$. The first equation to solve is the non linear integral equation for $\epsilon_0(\theta)$

$$\epsilon_0(\theta) = e^\theta - \int_{-\infty}^{\infty} \varphi(\theta - \theta') \log\left(1 + e^{-\epsilon_0(\theta')}\right) \frac{d\theta}{2\pi}. \quad (251)$$

We start by taking the finite set of real points $\{-aN, -a(N-1), \ldots, aN\}$ where $N \in \mathbb{N}$ and $a > 0$. For our numerics we set $N = 300$ and $a = 0.1$. We then discretise (251) so it becomes

$$\epsilon_0(ia) = e^{ia} - \frac{a}{2\pi} \sum_{j=-N}^{N} \varphi((i-j)a) \log\left(1 + e^{-\epsilon_0(ja)}\right), \quad i = -N, \ldots, N. \quad (252)$$

This discrete equation can then be solved iteratively. We take the seed solution

$$\epsilon_0^{(0)}(ia) = e^{ia}, \quad (253)$$

and then define $\epsilon^{(k+1)}(ia)$, for $k \geq 0$, by the recursion relation

$$\epsilon_0^{(k+1)}(ia) = e^{ia} - \frac{a}{2\pi} \sum_{j=-N}^{N} \varphi((i-j)a) \log\left(1 + e^{-\epsilon_0^{(k)}(ja)}\right), \quad (254)$$

We then evaluate a discrete version of the integral (110) giving the vacuum eigenvalue of $I_1$ using the solution $\epsilon_0^{(k)}$

$$L \mathcal{I}_1^{\text{vac},(k)}(L) = -\frac{a}{2\pi} \sum_{i=-N}^{N} e^{ia} L(\epsilon_0^{(k)}(ia)). \quad (255)$$

We terminate the algorithm when

$$\left| \frac{L\, \mathcal{I}_1^{\text{vac},(k+1)}(L) - L\, \mathcal{I}_1^{\text{vac},(k)}(L)}{L\, \mathcal{I}_1^{\text{vac},(k)}(L)} \right| < \delta \,, \tag{256}$$

for some chosen $\delta$. We set $\delta = 10^{-16}$ in our numerics and it typically took about 30 iterations before the iteration scheme terminated.

We now want to solve for the $\epsilon_n(\theta)$, $n \geq 1$, in the expansion (101). As remarked in (108) these all satisfy linear integral equations of the form

$$\epsilon_n(\theta) = f_n(\theta) - \int_{-\infty}^{\infty} \varphi(\theta - \theta')\epsilon_n(\theta')L'(\epsilon_0(\theta'))\frac{d\theta'}{2\pi} \,, \tag{257}$$

where the $f_n(\theta)$ is a know function of $\epsilon_0, \ldots, \epsilon_{n-1}$. We again discretise the integral and set up the iteration scheme for $k \geq 0$

$$\epsilon_n^{(k+1)}(ia) = f_n(ia) - \frac{a}{2\pi} \sum_{j=-N}^{N} \varphi((i-j)a)\epsilon_n^{(k)}(ja)L'(\epsilon_0(ja)) \,, \tag{258}$$

with seed solution

$$\epsilon_n^{(0)}(ia) = f_n(ia) \,. \tag{259}$$

We can again plug the solutions into a discrete version of the integral at $O(\alpha^n)$ in (109) to terminate the algorithm and find the desired energies.

## D.2 One Particle Excited State

We will now outline the numerical algorithm used to determine the excited states. We now have two equations to solve in tandem, one coming from the TBA equation (115) and the other coming from the constraint (117). We will start with the equations for $\epsilon_0$ and $\eta_0$, (122) and (127). The iteration scheme coming from (122) is

$$\epsilon_0^{(k+1)}(ia) = e^{ia} + \log\left(\frac{S(ia - \eta_0^{(k)})}{S(ia - \bar{\eta}_0^{(k)})}\right) - \frac{a}{2\pi} \sum_{j=-N}^{N} \varphi((i-j)a)L(\epsilon_0^{(k)}(ja)) \,, \tag{260}$$

for $k \geq 0$ with the seed solution

$$\epsilon_0^{(0)}(ia) = e^{ia} + \log\left(\frac{S(ia - \eta_0^{(0)})}{S(ia - \bar{\eta}_0^{(0)})}\right) \,. \tag{261}$$

In order to set up the iteration scheme for $\eta_0$ we first have to rearrange the constraint equation

$$2n\pi i = e^{\eta_0} - \log S(2i\text{Im}(\eta_0)) - \int_{-\infty}^{\infty} \varphi(\eta_0 - \theta')L(\epsilon_0(\theta'))\frac{d\theta'}{2\pi} \,. \tag{262}$$

We first rearrange it to put the $i\text{Im}(\eta_0)$ in $\log S(2i\text{Im}(\eta_0))$ on the left hand side

$$i\text{Im}(\eta_0) = \frac{1}{2}S^{-1}\left(\exp\left(e^{\eta_0} - \int_{-\infty}^{\infty} \varphi(\eta_0 - \theta')L(\epsilon_0(\theta'))\frac{d\theta'}{2\pi}\right)\right) \,. \tag{263}$$

$S^{-1}$ is the inverse of the S matrix which is given by

$$S^{-1}(\theta) = i \arcsin\left(\frac{\theta + 1}{\theta - 1}\sin\left(\frac{\pi}{3}\right)\right) \,. \tag{264}$$

The branch cut of arcsin is fixed by demanding that $\text{Im}(\eta_0) \in [0, 2\pi)$.

We can also extract the real part of $\eta_0$ by taking the imaginary part of (262). Taking the imaginary part gives

$$2n\pi = e^{\text{Re}(\eta_0)}\sin(\text{Im}(\eta_0)) - \pi s - \text{Im}\left(\int_{-\infty}^{\infty} \varphi(\eta_0 - \theta')L(\epsilon_0(\theta'))\frac{d\theta'}{2\pi}\right), \qquad (265)$$

where $s = 0$ if $S(2i\text{Im}(\eta_0)) > 0$ and $s = 1$ if $S(2i\text{Im}(\eta_0)) < 0$. This can be rearranged to give

$$\text{Re}(\eta_0) = \log\left(\frac{(2n+s)\pi + \text{Im}\left(\int_{-\infty}^{\infty} \varphi(\eta_0 - \theta')L(\epsilon_0(\theta'))\frac{d\theta'}{2\pi}\right)}{\sin(\text{Im}(\eta_0))}\right). \qquad (266)$$

Adding together (263) and (266) gives us the new constraint equation

$$\eta_0 = \log\left(\frac{(2n+s)\pi + \text{Im}\left(\int_{-\infty}^{\infty} \varphi(\eta_0 - \theta')L(\epsilon_0(\theta'))\frac{d\theta'}{2\pi}\right)}{\sin(\text{Im}(\eta_0))}\right) \qquad (267)$$
$$+ \frac{1}{2}S^{-1}\left(\exp\left(e_0^{\eta} - \int_{-\infty}^{\infty} \varphi(\eta_0 - \theta')L(\epsilon_0(\theta'))\frac{d\theta'}{2\pi}\right)\right).$$

Through numerical experimentation we found that this appears to be best form of the constraint equation to turn into an iteration scheme. Our discrete iteration scheme is then

$$\eta_0^{(k+1)} = \log\left(\frac{(2n+s)\pi + \text{Im}\left(\frac{a}{2\pi}\sum_{i=-N}^{N}\varphi(\eta_0^{(k)} - ia)L(\epsilon_0^{(k)}(ia))\right)}{\sin(\text{Im}(\eta_0^{(k)}))}\right) \qquad (268)$$
$$+ \frac{1}{2}S^{-1}\left(\exp\left(e^{\eta_0^{(k)}} - \frac{a}{2\pi}\sum_{i=-N}^{N}\varphi(\eta_0^{(k)} - ia)L(\epsilon_0^{(k)}(ia))\right)\right).$$

We set the initial value of $\eta_0^{(0)} = 2.2 + 0.5i$ and found that our scheme converges to the correct solutions in about 30 iterations. We then solve (260) and (268) in tandem. The solutions can then be plugged into a discretisation of the $O(\alpha^0)$ integral in (131) to determine the excited states.

In order to solve for $\epsilon_n$ and $\eta_n$ for $n \geq 1$ we have the TBA equation (126)

$$\epsilon_n(\theta) = g_n(\theta) + 2\text{Im}(\eta_n\varphi(\theta - \eta_0)) - \int_{-\infty}^{\infty} \varphi(\theta - \theta')\epsilon_n(\theta')L'(\epsilon_0(\theta'))\frac{d\theta'}{2\pi}, \qquad (269)$$

and the constraint (130)

$$0 = h_n + \eta_n e^{\eta_0} + 2\text{Im}(\eta_n)\varphi(2i\text{Im}(\eta_0)) - \int_{-\infty}^{\infty}(\eta_n\varphi'(\eta_0 - \theta)L(\epsilon_0(\theta)) + \epsilon_n(\theta)\varphi(\eta_0 - \theta)L'(\epsilon_0(\theta)))\frac{d\theta}{2\pi}, \qquad (270)$$

where $g_n(\theta)$ and $h_n$ depend on $\epsilon_i$ and $\eta_i$ for $i = 1, \ldots, n-1$ which have previously been determined. As we did above we will rearrange the constraint equation before setting up an iterative scheme. The imaginary part of $\eta_n$ can be solved for to give

$$\text{Im}(\eta_n) = \frac{1}{2\varphi(2i\text{Im}(\eta_0))}\left(\int_{-\infty}^{\infty}(\eta_n\varphi'(\eta_0 - \theta)L(\epsilon_0(\theta)) + \epsilon_n(\theta)\varphi(\eta_0 - \theta)L'(\epsilon_0(\theta)))\frac{d\theta}{2\pi} - h_n - \eta_n e^{\eta_0}\right). \qquad (271)$$

1047 To get the real part of $\eta_n$ we take the imaginary part of (270) and rearrange

$$\mathrm{Re}(\eta_n) = \frac{\mathrm{Im}\left(\int_{-\infty}^{\infty}(\eta_n\varphi'(\eta_0-\theta)L(\epsilon_0(\theta))+\epsilon_n(\theta)\varphi(\eta_0-\theta)L'(\epsilon_0(\theta)))\frac{d\theta}{2\pi}-h_n\right)-\mathrm{Im}(\eta_n)\mathrm{Re}(e^{\eta_0})}{\mathrm{Im}(e^{\eta_0})}.$$
(272)

1048 Taking the sum of these two expressions we find a new form of the constraint

$$\eta_n = \frac{\mathrm{Im}\left(\int_{-\infty}^{\infty}(\eta_n\varphi'(\eta_0-\theta)L(\epsilon_0(\theta))+\epsilon_n(\theta)\varphi(\eta_0-\theta)L'(\epsilon_0(\theta)))\frac{d\theta}{2\pi}-h_n\right)-\mathrm{Im}(\eta_n)\mathrm{Re}(e^{\eta_0})}{\mathrm{Im}(e^{\eta_0})}$$
$$+ \frac{i}{2\varphi(2i\mathrm{Im}(\eta_0))}\left(\int_{-\infty}^{\infty}(\eta_n\varphi'(\eta_0-\theta)L(\epsilon_0(\theta))+\epsilon_n(\theta)\varphi(\eta_0-\theta)L'(\epsilon_0(\theta)))\frac{d\theta}{2\pi}-h_n-\eta_ne^{\eta_0}\right).$$
(273)

1049 We take a discrete version of this constraint and a discrete version of (269) to obtain the two
1050 recursive equations

$$\epsilon_n^{(k+1)}(ia) = g_n(ia) + 2\mathrm{Im}(\eta_n^{(k)}\varphi(ia-\eta_0)) - \frac{a}{2\pi}\sum_{j=-N}^{N}\varphi((i-j)a)\epsilon_n^{(k)}(ja)L'(\epsilon_0(ja)), \quad (274)$$

1051 with

$$\epsilon_n^{(0)}(ia) = g_n(ia) + 2\mathrm{Im}(\eta_n^{(0)}\varphi(ia-\eta_0)), \quad (275)$$

1052 and

$$\eta_n^{(k+1)} = \frac{\mathrm{Im}\left(\frac{a}{2\pi}\sum_{i=-N}^{N}\left(\eta_n^{(k)}\varphi'(\eta_0-ia)L(\epsilon_0(ia))+\epsilon_n^{(k)}(ia)\varphi(\eta_0-ia)L'(\epsilon_0(ia))\right)-h_n\right)-\mathrm{Im}(\eta_n^{(k)})\mathrm{Re}(e^{\eta_0})}{\mathrm{Im}(e^{\eta_0})}$$
$$+ \frac{i}{2\varphi(2i\mathrm{Im}(\eta_0))}\left(\frac{a}{2\pi}\sum_{i=-N}^{N}\left(\eta_n^{(k)}\varphi'(\eta_0-ia)L(\epsilon_0(ia))+\epsilon_n^{(k)}(ia)\varphi(\eta_0-ia)L'(\epsilon_0(ia))\right)-h_n-\eta_n^{(k)}e^{\eta_0}\right).$$
(276)

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
