# Peer review of "Modular Properties of Generalised Gibbs Ensembles"

_SciPost Physics_

## Round 1 · Referee Report · Anonymous (Referee 1) · 2025-1-3

Strengths
-The paper addresses a timely problem
-The paper contains original and rigorous results
-The paper cites the relevant literature
-The paper points to further developments and open problems
Weaknesses
Report
The authors address this problem by employing several methods, notably, the thermodynamic Bethe ansatz approach and also by interpreting the GGE partition function, as the partition function of a Gibbs ensemble in the presence of a conformal defect.
These methods are then exemplified for a the simplest non-trivial non-unitary minimal model, the Lee-Yang CFT where in particular the TBA can be solved perturbatively in the higher spin charge coupling. It is conjectured that the TBA description of the GGE is complete in the sense that it encodes information about the averages of all higher conserved charges.
As I wrote elsewhere, this paper has many strengths and the potential weakness of containing many analytical and numerical results which are highly model specific. However, the methodology is rather general and the intention to generalise and extend the results is stated in the conclusion. It is my view that the paper deserves to be published in SciPost in its present form.
Requested changes
The paper is well written and I think that it could be published in its current form. I would like to suggest two minor changes.
1) I suggest citing the paper https://arxiv.org/abs/1203.1305 since it contains an early derivation of the TBA equations for GGEs. The fact that the driving term of the TBA should be modified by including all conserved quantities had already been pointed out in https://arxiv.org/pdf/0911.3345
2) I am not 100% sure about this since there are already a lot of Appendices in this paper. However, I would suggest that perhaps parts of subsections 5.4 and 5.5 could be moved to an Appendix too (some of the longer formulae and tables) and that this would make this section more readable.
Recommendation
Publish (easily meets expectations and criteria for this Journal; among top 50%)
We are grateful to the referee of Report 1 on 2025-1-3 for their close reading and constructive & thoughtful comments. Considering the feedback we have received; we will update our paper and hope that these changes are sufficient:
1) We will appropriately cite the two papers that were suggested to us.
2) We have decided not to move the subsections 5.4 and 5.5 into the Appendix as we feel that (a) in agreement with the referee that we already have many Appendices and (b) we wanted to make more apparent in the main text exactly how we went about justifying how the modular transformation of the GGE is encoded in the TBA. While we agree that there are formulae in these sections that are long and cumbersome, we are not necessarily convinced that this would improve readability since a reader would have to go back and forth between the appendix and the main text.

Author: Faisal Karimi on 2025-02-10 [id 5205]
(in reply to Report 4 on 2025-02-09)We would like to thank the referee of Report #4 for their kind words and careful reading of our paper.
It seems that no requested changes have been made in this report, and so we will make appropriate changes according to all of the feedback from all of the referees and resubmit a new version, as has now been asked for in the editorial recommendation.

---

## Round 1 · Referee Report · Anonymous (Referee 2) · 2025-2-1

Strengths
Weaknesses
Report
The paper is very detailed, it contains a number of interesting calculations, relying on different techniques (including TBA, which to my knowledge, was not applied to this problem before). I think results of this paper will be useful for future studies of the KdV GGE in 2d CFTs.
Requested changes
Some of the calculations (in Appendix B) of the zero modes of the quasi-primary fields on the cylinder were discussed in the literature before. For example eq. (211) is the same as eq. (4.4) in 1912.13444 (there are also other examples). It would be proper to acknowledge the overlap.
Recommendation
Publish (easily meets expectations and criteria for this Journal; among top 50%)
We would like to thank the referee for taking the time to carefully read the paper and provide useful comments on the manuscript. In regard to their comments we agree that it was an oversight for us not to cite 1912.13444 and will appropriately sight it in the paper (along with the additional citations provided by referee 1).
Strengths
Weaknesses
Report
I think the paper can be published in its current form.
Recommendation
Publish (easily meets expectations and criteria for this Journal; among top 50%)

---

## Round 1 · Referee Report · Anonymous (Referee 3) · 2025-2-9

Report
The paper studies two dimensional conformal field theory (CFT), which is known to have an infinite number of mutually commuting higher spin conserved charges $I_n$, which are called the quantum KdV charges. The main focus of the work is the modular property of the Generalized Gibbs Ensemble (GGE), in which these higher spin charges are inserted in the toroidal partition function. Most of the work is restricted to the Lee-Yang model, in which the fugacity $\alpha$ for a single charge $I_5$ is turned on, though there are some results also for GGEs in a Verma module. This work can be considered an extension and generalization of earlier work by one of the authors in [16,17], in which the question of modularity of the GGE was addressed in the context of the free fermion model.
In the first part of the paper, the GGE is expanded for small fugacity in an asymptotic series. Then, the first few terms essentially amounts to the calculation of thermal correlators of the form $\langle I_5^n \rangle$, where the angular brackets indicate the thermal vev in the vacuum module. For small $n$, these are calculated along with their $S$-transforms. The left hand side can be written as quasimodular differential operators acting on the characters of the model, and their S-transforms can be computed easily. These are then repackaged as linear combinations of correlators of quasiprimaries. A useful technical point here is the use of a basis that already takes into account the null vectors within the module of the minimal model, that leads to the evaluation of the vevs and their S-transforms up to $O(\alpha^3)$. The work done by the authors builds upon earlier literature on thermal correlators [7] and also the work specific to the Lee-Yang model in [40]. This is already a non-trivial calculation, and a rather striking result is obtained: the S-transform of the GGE involves not only higher KdV charges but also zero-modes of quasi-primary fields $J_k$ that are conserved but that are not in involution with the KdV charges $I_n$. This is a novel result.
The authors then propose to resum this asymptotic expansion into a transformed GGE, in which both the quantum KdV charges and the zero-modes of the quasi-primary operators are included. This is a formal sum of an infinite number of terms that diverges, which therefore needs regularization. This leads to the second part of the project in which the spectrum of the transformed GGE is obtained by means of the thermodynamic Bethe ansatz. This part is quite technical and there is an impressive amount of numerical work that is done to provide evidence for this proposal. The fundamental claim here is that if the spectrum of the original GGE (with non-vanishing fugacity for a single quantum KdV charge) is obtained by a TBA (and this has been known since the seminal work in [24]), then the spectrum of the S-transformed GGE (which includes non-commuting charges) can be obtained from the mirror-TBA of the Lee-Yang model. The resulting analysis is new and the authors clearly explain how to systematically compute the spectrum using asymptotic solutions to the mirror TBA.
The last point that is made is to show the existence of non-asymptotic solutions to the TBA. Based on previous and analogous work done for the free fermion model, this leads to a conjecture that terms arising from these non-asymptotic solutions also have to be added in order to find the full modular transformation of the original GGE. This last part is conjectural, with more work needed to prove the claims made here, but the authors are aware of this.
This is a well written paper that deserves to be published. The questions that they raise, namely the modular transformation of GGEs, is a worthwhile one to explore. The answers that they provide, using the analysis of the TBA for the Lee-Yang model, are compelling.
Recommendation
Publish (easily meets expectations and criteria for this Journal; among top 50%)

---

## Editorial Decision

unknown